# Optimizing Radar Scan Strategies for Tracking Isolated Deep Convection Using Observing System Simulation Experiments

Mariko Oue[1], Stephen M. Saleeby[2], Peter J. Marinescu[2,4], Pavlos Kollias[1,3], and
Susan C. van den Heever[2]

[1.]School of Marine and Atmospheric Sciences, Stony Brook University, Stony Brook, NY, USA
[2.]Department of Atmospheric Science, Colorado State University, Fort Collins, CO, USA
[3.]Environmental and Climate Sciences Department, Brookhaven National Laboratory, Upton, NY, USA
[4.]Cooperative Institute for Research in the Atmosphere, Colorado State University, Fort Collins, CO, USA

*Correspondence to*: Mariko Oue (mariko.oue@stonybrook.edu)

**Abstract.** Optimizing radar observation strategies is one of the most important considerations in pre-field campaign periods. This is especially true for isolated convective clouds that typically evolve faster than the observations captured by operational radar networks. This study investigates uncertainties in radar observations of the evolution of the microphysical and dynamical properties of isolated deep convective clouds developing in clean and polluted environments. It aims to optimize the radar observation strategy for deep convection through the use of high spatiotemporal cloud-resolving model simulations, which resolve the evolution of individual convective cells every 1 minute, coupled with a radar simulator and a cell tracking algorithm. The radar simulation settings are based on the Tracking Aerosol Convection Interactions ExpeRiment (TRACER) / Experiment of Sea Breeze Convection, Aerosols, Precipitation and Environment (ESCAPE) field campaigns held in the Houston, TX area, but are generalizable to other field campaigns focusing on isolated deep convection. Our analysis produces the following four outcomes. First, a 5-7 m s$^{-1}$ median difference in maximum updrafts of tracked cells is shown between the clean and polluted simulations in the early stages of the cloud lifetimes. This demonstrates the importance of obtaining accurate estimates of vertical velocity from observations if aerosol impacts are to be properly resolved. Second, tracking of individual cells and using vertical cross section scanning every minute captures the evolution of precipitation particle number concentration and size represented by polarimetric observables better than the operational radar observations that update the volume scan every 5 minutes. This approach also improves the multi-Doppler radar updraft retrievals above 5 km above ground level for regions with updraft velocities greater than 10 m s$^{-1}$. Third, we propose an optimized strategy which is composed of cell tracking by quick (1-2 min) vertical cross section scans from more than one radar in addition to the operational volume scans. We also propose the use of a single range-height indicator updraft retrieval technique for cells close to the radars, where the multi-Doppler radar retrievals are still challenging. Finally, increasing the number of deep convective cells sampled by such observations better represents the median maximum updraft

evolution with sample sizes of more than 10 deep cells, which decreases the error associated with sampling the true
population to less than 3 m s$^{-1}$.
**1 Introduction**
The quality and performance of remote sensing measurements, especially radar measurements, can strongly depend on
the siting of instruments relative to their targets and the associated sampling strategies (e.g., Bousquet et al., 2008; Potvin
et al., 2012b; Oue et al., 2019). This is especially true for convective storm systems that evolve rapidly over a range of
spatial and temporal scales. The limitations associated with observation strategies influence microphysical, dynamical, and
convective-core property retrievals, resulting in a misinterpretation of the observational data and can limit our understanding
of storm processes. Some of these limitations can be addressed using Model and field Experiment data fusion (ModEx)
concepts such as the optimization of experimental design using models and forward simulators. Using the ModEx
framework, one can appropriately determine optimal radar deployments and scan strategies, as well as quantitatively
understand the observational uncertainties arising from these strategies before field campaigns begin. As such, the goal of
this study is to suggest optimal radar deployments and scan strategies for radar field campaigns targeting isolated convective
clouds.
The limitations in the radar observations that are mostly attributed to sampling strategy strongly impact the radar-based
retrievals of geophysical quantities and cloud properties (e.g., Clark et al., 1980; Given and Ray, 1994; Collis et al., 2010).
These sampling strategy choices include scanning time (scan rate) for a volume scan, spatial resolution (azimuth/elevation
spacings and range-gate spacing), elevation angles for plan position indicator (PPI) volume scans, distance to the target
phenomena from the radars (this is also related to the spatial resolution), systematic variability (i.e., noisiness) in the
observables, and data smoothing and interpolation for gridding the data. In particular, the sampling strategy can significantly
impact the uncertainties in vertical velocity retrievals (e.g., Oue et al., 2019), which are important for the analysis of cloud
microphysics and dynamics. In addition, the retrievals include uncertainties attributed to assumptions in their algorithms;
for instance, some multi-Doppler radar-based vertical velocity retrievals must make assumptions for the particle fall speed
and mass continuity (e.g., North et al., 2017).
In operational radar networks (e.g., the Next Generation Weather Radar (NEXRAD) network), each radar performs
volume scans consisting of plan position indicator (PPI) scans with multiple elevation angles to prioritize collecting data
for large areas. The volume scan strategy (known as volume coverage pattern, VCP) takes approximately 5 minutes to
collect the 3D atmospheric data. While this operational scanning strategy is very valuable for performing surveillance and
collecting a large number of cloud samples, it may not accurately capture fine-scale, rapidly-developing cloud phenomena.
To increase our understanding of the links between convective cloud kinematic and microphysical processes, field
campaigns have recently started to focus on collecting observations at higher temporal and spatial resolutions to understand
fine-scale characteristics and phenomena including isolated convection, shallow cumulus clouds, plumes embedded in
mesoscale systems, convective updrafts and downdrafts (e.g., Verification of the Origins of Rotation in Tornadoes

Experiment 2 (VORTEX2), Wurman et al., 2012; Midlatitude Continental Convective Clouds Experiment (MC3E), Jensen et al., 2016; CSU Convective Cloud Updraft and Downdraft Experiment (C³LOUD-Ex), van den Heever et al., 2021; Marinescu et al., 2020), and high-latitude precipitation (e.g., Light Precipitation Evaluation Experiment (LPVEx), L'Ecuyer, et al., 2010). Furthermore, in some of these field campaigns, physically tracking individual convective phenomena using cutting-edge mechanically scanning radar systems was employed to prioritize high spatiotemporal sampling (e.g., The Dynamical and Microphysical Evolution of Convective Storms (DYMECS), Stain et al., 2015; Iowa Flood Studies (IFloodS), Mishra et al., 2016). The high-spatiotemporal resolution observations can also be achieved by complementing the operational radar networks with adapting scan strategies of the regional research radars that have been installed in local areas. (e.g., Distributed Collaborative Adaptive Sensing (DCAS), McLaughlin et al., 2005; Multi-sensor Agile Adaptive Sampling (MAAS), Kollias et al., 2020).

In recent years, as phased array weather radars (PARs) have become more commonly used for severe weather observations, the sophisticated tracking of atmospheric phenomena has become feasible (e.g., Kollias et al., 2022). The PARs have a significant advantage of sampling rapidly evolving atmospheric phenomena at high-temporal resolutions (e.g., Billam and Harvey, 1987; Heinselman and Torres, 2011; Mahre et al., 2018; Griffin et al., 2019; Adachi and Mashiko, 2020; Moroda et al., 2021), thus allowing for sampling of the entire cloud volume and cloud lifecycle. The tracking observations obtained by these rapid scanning radar or PAR systems are, however, more sensitive than previous approaches to scan strategies such as sampling time, azimuth/elevation spacings, and deployments (locations and the number of radars), all of which should be appropriately optimized depending on the spatial scale and evolution speed of the target phenomena (Kollias et al., 2020).

Several radar field campaigns aim at enhancing our understanding of the links between convective cloud kinematic and microphysical processes and life cycles (e.g., Tracking Aerosol Convection Interactions ExpeRiment (TRACER), Jensen et al., 2019; Experiment of Sea Breeze Convection, Aerosols, Precipitation and Environment (ESCAPE); Jensen et al., 2022; Atmospheric Radiation Measurements (ARM) Mobile Facility 3 (AMF3) Southeast US deployment, Kang et al., 2021). All of these experiments have deployed or plan to deploy multiple mobile weather radars, cloud radars, rapid-scan radars, and phased array radars. In particular, TRACER and ESCAPE campaigns focus on observing isolated deep convective storms with different aerosol environments. Optimizing the radar deployments and scan strategies while taking into account campaign costs, deployment limitations, and sampling limitations (i.e., range, scan rate) is a large but critical challenge. Observing system simulation experiment (OSSE) is a powerful tool to investigate the impact of the limitations on the observation analyses (Oue et al., 2020), and using high-spatiotemporal data is needed to reliably simulate the observations accounting for the limitations. While the focus of this study has been on the TRACER/ESCAPE field campaigns, the results are generalizable to other future campaigns focused on isolated deep convection. In this paper we make use of OSSEs focused on deep convection to specifically investigate the impacts of radar scan strategies on the cell tracking performance, microphysical evolution, and dynamical retrievals of convective storms. Specifically, the impacts of varying the scan elevation angles, the period for a volume scan, and the locations of the radars are assessed.

## 2 Method

Our OSSE approach is comprised of three parts: (1) the Regional Atmospheric Modeling System (RAMS; Cotton et al., 2003; Saleeby and van den Heever, 2013); (2) the Cloud-resolving Radar Simulator (CR-SIM; Out et al., 2020); and (3) the Tracking and Object-Based Analysis of Clouds (*tobac*; Heikenfeld et al., 2019; Sokolowsky et al., 2022). Figures 1a-1c show example snapshots from parts (1) and (2), and Figure 1d shows the tracking result from the part (3). RAMS model output from the Aerosol-Cloud-Precipitation-Climate (ACPC) model intercomparison project (MIP) (van den Heever et al., 2018; Marinescu et al., 2021), which focuses on the development and occurrence of isolated convective cells in the region around Houston, TX, on June 19-20, 2013 (Fig. 1a), forms the basis of this study. The convective development was initiated both along the inland propagation of the sea breeze, and later in association with convective cold pools produced by the earlier convection in the simulation. In this study we focus on the time period from 20-24 UTC (15-19 local time) during which deep convective clouds developed, the dynamical processes of which have been extensively analyzed (Marinescu et al., 2021). One-minute simulated deep convective fields are used as an input to CR-SIM to represent and evaluate the radar observable fields (Fig. 1b). The CR-SIM radar observables are subsequently used to track convective cells using *tobac*.

### 2.1 CR-SIM

CR-SIM is a sophisticated radar forward operator developed to bridge the gap between high-resolution cloud model output and radar observations (Oue et al., 2020). CR-SIM can be applied to the 3D model output produced by a variety of cloud-resolving models and large-eddy simulation models, including RAMS, the Weather Research and Forecasting (WRF, Powers et al., 2017) model, the System for Atmospheric Modeling (SAM, Khairoutdinov and Randall, 2003), Cloud Model 1 (CM1, Bryan and Fritsch, 2002), and the Icosahedral Nonhydrostatic model (ICON, Zängl et al., 2015). It emulates the interaction between transmitted polarized radar waves and rotationally symmetric hydrometeors and can simulate the power (equivalent radar reflectivity factor), phase (Doppler velocity), and polarimetric (specific differential phase, differential reflectivity, depolarization) variables with either a fixed elevation angle or varying elevation angles with respect to a specified radar location. CR-SIM outputs these variables on the same grid as the input model grid. The radar simulator has been shown to be especially effective in OSSEs to investigate the uncertainties in observational data (Oue et al., 2019).

### 2.2 *tobac*

*tobac* is a python-based software platform specifically developed for tracking atmospheric features, such as isolated convective cells, in both model and observational datasets. *tobac* has been developed using a modular code structure with data input, feature detection and segmentation, and trajectory linking steps. It uses a watershed algorithm to detect and track individual convective cells, and it has been extensively tested on the ACPC simulations (e.g., Heikenfeld et al., 2019;

Marinescu et al., 2021). For this study, *tobac* Version 1.2 is applied to CR-SIM vertically integrated liquid (VIL; Fig. 1c),
which represents the total hydrometeor condensate within each vertical column and is similar to the approach used by Hu
et al. (2019). The CR-SIM radar reflectivity is converted into VIL using the following equation:
$VIL = \sum_{i=0}^{i=imax} 3.44 * 10^{-6}[(Z_i + Z_{i+1})/2]^{4/7}(h_{i+1} - h_i) \quad [kg\ m^{-2}]$         (1)
where Z is radar reflectivity factor (mm$^6$ m$^{-3}$), $h$ is height (m), $i$ is the vertical index, and $imax$ is the index at the grid
domain top. We calculate VIL using the CR-SIM-simulated total reflectivity greater than or equal to 0 dBZ at all vertical
levels and, thus, ensure that we consider all cloudy grid boxes in the tracking analysis. Although this variable is named
'liquid,' we use the total reflectivity from all simulated hydrometeor species to emulate real observations, including cloud
droplets, drizzle, rain, cloud ice, snow, aggregates, graupel, and hail. Since "VIL" is a widely-used name, we refer to VIL
as this parameter. When considering that clouds may have lower reflectivity (< 0 dBZ) and the radar minimum detectable
reflectivity increases with distance from the radar, the reflectivity threshold of 0 dBZ for the VIL calculation is a reasonable
value to use in detecting cells in the entire domain regardless of the distance. We also performed the cell tracking using 10
and 40 dBZ thresholds at the height of 2 km above ground level (AGL) to compare the performance of the use of VIL and
single-level reflectivity thresholds.
**2.3 RAMS**

RAMS is a cloud-resolving model that includes sophisticated microphysical-dynamical feedbacks, as well as aerosol-

cloud interactions (Saleeby and van den Heever, 2013). RAMS, along with several other cloud resolving models from
around the world participated in the ACPC MIP, focuses on the effects of changing the concentrations of cloud condensation
nuclei (CCN) on deep convective clouds (van den Heever et al., 2018). Case study simulations of a period of scattered
convective clouds near Houston, Texas were completed with relatively low and high concentrations of CCN that were based
on observations from the Houston area (see Figure 2 from Marinescu et al., 2021). The low-CCN simulation is initialized
with 500 cm$^{-3}$ of CCN in the boundary layer (named CLN in this study), while the high-CCN simulation is initialized with
4000 cm$^{-3}$ of CCN in the boundary layer (named POL in this study). The vertical aerosol profiles of both the CLN and POL
studies decrease linearly from the top of the boundary layer to 150 cm$^{-3}$ at ~5 km AGL (the free troposphere), above which
they remain constant. RAMS allows for the advection, nucleation, wet and dry deposition, and regeneration of aerosol
particles via hydrometeor evaporation and sublimation. These simulations have been performed using a horizontal grid
resolution of 500 m and RAMS' two-moment bin-emulating bulk-microphysics scheme, which predicts the mass and
number of eight hydrometeor types. The model data are output at a frequency of 1-minute. Additional details about the
RAMS model parameterizations and experimental setup used for these simulations can be found in Table 1 of Marinescu et
al. (2021).

## 2.4 Observation simulation processes

In this study, the cell tracking is applied to the CR-SIM-simulated radar observation field (VIL) to detect and track individual convective storm cells. Using the tracking results for all cells, we investigate the performance of the cell tracking using VIL, the impact of the scan strategy on the VIL estimates (Sect. 3.1), and the statistical impact of aerosols on the cell dynamical evolution (Sect. 3.3). One of the tracked, isolated, deep convective cells with a single precipitation core is chosen to investigate the following: 1) the impacts of the scan strategy on the examination of polarimetric observables and related microphysical studies (Sect. 3.2); and 2) the influences of different sets of the scan strategies on the multi-Doppler vertical velocity retrievals (Sect. 3.3). The chosen cell is representative of isolated deep convective cells from the CLN simulation (discussed in Sect. 3.2).

### 2.4.1. Tracking convective cells

The *tobac* cell tracking is coupled with CR-SIM radar observables obtained using the RAMS model output in the following manner:

1) The RAMS model output from the ACPC MIP for an isolated convective case over the Houston area (Fig. 1a) is used as input to CR-SIM.

2) The radar observable fields (Fig. 1b) are simulated using CR-SIM and output on the same grid as the input model grid.

3) The CR-SIM simulated radar reflectivity is converted into VIL if the reflectivity exceeds 0 dBZ at all levels (Fig.1c).

4) *tobac* is applied to the VIL field to track the convective cells (Fig. 1d). We used the VIL thresholds of 0, 0.1, 1.0, and 5.0 kg m$^{-2}$ to identify/track individual cells, including those embedded in larger precipitation areas.

5) Steps 1-4 above are applied to the CLN and POL RAMS simulations to investigate the impact of aerosols on the cell dynamical evolutions in the entire simulation domain.

### 2.4.2. Emulating radar scan strategies

We emulate the radar scan strategies to account for observational limitations including scanning time for a volume scan, azimuth/elevation angle spacings, range-gate spacing, elevation angles for PPI volume scans, distance to the target phenomena from the radars, and smoothness and interpolation for gridding process. This study tests the sensitivity of updraft retrievals to four of these scanning strategy choices: i) scanning time, ii) elevation angle spacing, iii) distance to the targeted convective cell, and iv) the number of radars used for the updraft retrievals. This section explains how the scan strategies are emulated using CR-SIM.

The various radar scan strategies emulated in this study are listed in Table 1. We first emulate cell tracking using sector range-height indicator (RHI) scans, each of which is composed of full elevation angles from 0.5° to 89.5° with a 1° increment

in an azimuth sector and takes approximately 1 minute (1-min RHI in Table 1). The 1-min RHI scan uses a snapshot of data to complete a full elevation scan for a sector. For mechanically scanning radars, 1-min RHI may not be feasible due to mechanical limitations (e.g., overhead time needed when changing the antenna sweep direction), and those radars may need more time to complete the sector scans (as discussed later in this section).

The second emulation of cell tracking is also a full elevation scan for an azimuth sector similar to 1-min RHI but takes 2 minutes using two continuous snapshots (2-min SEC). To construct the cells observed by the sector scan that takes 2 min, we use two consecutive model snapshots; the first snapshot at the earlier time is used to simulate the scan for angles from $0.5°$ to $44.5°$ over the elevation, and the other is used to simulate the scan for the angles from $45.5°$ to $89.5°$ over the elevation (we intend this simulation to represent a 2-min "RHI" in which each of the two snapshots should be used for a half of the azimuth sector for full elevation angles, however, for technical and computational reasons, we separate the elevation angles into the two snapshots). This 2-min SEC simulation is performed every 2 min.

The tracking cell by 1-min RHI and 2-min SEC is guided by *tobac* using the VIL estimate from the model full grid every 1 minute. The azimuth sectors for 1-min RHI and 2-min SEC are decided so that each azimuth sector covers the 10-km width centered around the individual cells defined by *tobac*. Therefore, the number of RHI sweeps for each cell varies as a function of the distance between the radar and the target cell. The radar configuration for the RHI simulation is assumed to be a general scanning radar such as the ARM precipitation radars. The angle range for an azimuth sector at the radar range of 40 km is approximately $14°$. With the radar beam width of $1°$, the total beam for the sector scan is 90 (over elevation) x 14 (over azimuth) = 1260 beams. Assuming that each beam uses ~96 radar pulse samples, the sector scan includes 120960 pulses in total. If the radar operates with 1.5 KHz pulse repetition frequency (PRF) (typical value for C-band radars), then the sector scan takes 80 sec; and if the radar operates with 2.5 kHz PRF (typical value for X-band radars), then the scan takes 48 sec. These numbers (scans within 1-2 min) are easy to get for phased-array radar observations. For a reflector (mechanical scan) radar that needs 33% overhead time due to acceleration and deceleration of the antenna, these scan times become 106 sec and 64 sec respectively.

The third strategy we investigate is the 5-min VCP. This strategy follows the standard NEXRAD VCP precipitation mode (VCP 12, https://www.weather.gov/jetstream/vcp_max) and is composed of 14 PPI scans. Since our model output is every minute, for the 5-min VCP simulation, a volume scan is composed of 5 snapshots from the 1-min model outputs. A single snapshot is used to create two or three PPI sweeps (two or three elevation angles).

Finally, for the fourth strategy, we evaluate an "ideal" simulation where a volume scan with full elevation and azimuth scans with a $1.0°$ increment over both elevation and azimuth is performed within 1-min (referred to as "Full" in Table 1). This approach will be feasible when a network of rapid scan or electronically scanning radars is available. Although such observations are not realistic, they can serve as an upper boundary in terms of observational capabilities and will be used for an evaluation of VIL from 5-min VCP in Section 3.1.

We use an S-band frequency for the 5-min VCP simulation (emulating NEXRAD radars) and a C-band frequency for 1-min RHI, 2-min SEC, and Full simulations (assuming the C-band Scanning ARM Precipitation Radar (C-SAPR), or any

equivalent performance radar). Since we use unattenuated radar observables in this study, the impacts of the radar frequency
on the simulation results should not be significant.

### 2.4.3. Multi Doppler radar wind retrieval

For the investigation of the impacts of scan and deployment strategies on multi-Doppler vertical velocity retrievals, this
study employs a three dimensional variational (3DVAR) multi-Doppler radar wind retrieval technique developed by North
et al. (2017). While this investigation focuses on uncertainties caused by scan and deployment strategies, it does not account
for other sources of errors such as attenuation, nor the particle fall speed assumed in the 3DVAR wind retrieval technique.
We use unattenuated radar reflectivity and reflectivity-weighted fall speed calculated by CR-SIM in all present wind
retrieval simulations. The details of the 3DVAR retrieval settings are presented in Oue et al. (2019). As described in Oue et
al. (2019), the 3DVAR wind retrieval technique is applied to the gridded radar observable fields. The radar observables that
are resampled following the radar scan strategies in the previous sections are then re-gridded into a Cartesian coordinate of
250 km x 250 km x 14 km domain with 0.25-km horizontal and vertical spacings using Barnes distance-dependent
weightings (Barnes, 1964).

### 3. Results


### 3.1 Evaluation of the tracking parameter

This study employs VIL as a tracking parameter and, as such, is similar to Hu et al. (2019). The use of VIL allows us
to consider hydrometeor condensate at all levels, whereas previous convective cell tracking studies have employed
reflectivity criteria at a given height (e.g., Steiner et al., 1995; Shusse et al., 2006; Oue et al., 2014). Tracking based on
reflectivity at a single height may well define individual cells especially for embedded cells in stratiform regions, however,
it can miss some of the early stages of convective cell development that initiate at different (typical lower) heights. In this
section, we evaluate VIL as a tracking parameter for the simulations used in this study. Figure 2a shows a comparison of
the durations of *tobac* detected and tracked cells in the CLN simulation as a function of the use of VIL, as well as 10 and
40 dBZ thresholds at 2 km altitude. The time bin size used for Fig. 2 is 5 min. The VIL-based tracking has the largest total
number of cells detected since the VIL better captures the presence of hydrometeor condensate throughout the vertical
columns and is not dependent on the presence of condensate at a specific level. All of the frequency distributions, perhaps
unsurprisingly, peak at shorter durations for both CLN and POL cases. The VIL-based and 10-dBZ-based tracking are more
comparable, although the VIL-based tracking has higher frequencies at even longer durations (> 90 min) compared to the
10-dBZ-based tracking. The 40-dBZ-based tracking generally has lower frequencies at all duration time bins compared to
the 10-dBZ- and VIL-based tracking, but it is more similar to the 10-dBZ-based tracking in the 25–40 minute time bins.
The frequency distributions of tracked cell lifetimes suggest that VIL can better capture longer life cycles of individual cells,

including their initial development and decay stages, due to its ability to include information about hydrometeors in the entire column.

The POL simulation (dashed line in Figure 2a) shows a similar tracked cell lifetime distribution to the CLN case. However, there are some notable differences. The POL case has fewer cells detected (~15% fewer for VIL), which is consistent with Marinescu et al. (2021), who also found fewer deep convective updrafts in the POL case using different analyses (their Figure 7). When considering the relative frequency distribution (not shown), the POL case also has a distribution shift towards relatively fewer long-lived cells (lifetimes > 20 mins) and more frequent short-lived cells (lifetimes < 20 mins), as compared to the CLN case. The relatively fewer long-lived cells in the POL case are associated with deep convection. There could be several reasons for the difference in cell lifetimes related to microphysical-dynamical feedback processes, such as those associated with cold pools (e.g., Grant and van den Heever, 2015). These differences between CLN and POL are being examined in a separate manuscript. Hereafter, we use the CLN case to examine the effects of scan strategy on the radar polarimetric observables and vertical velocity retrievals. The difference in the number of cells detected is consistent between the three tracking criteria. However, the difference in lifetime is clearest in VIL, being slightly evident in 10-dBZ-based, but unclear in 40-dBZ-based. This suggests that the VIL-based tracking is more sensitive to the difference in cell lifetimes between the CLN and POL simulations, and therefore, may be suitable for tracking isolated convective cells throughout their lifetimes and quantifying cell-lifetime statistics. This may work for the cases where isolated cells dominate in the domain with less stratiform or mesoscale precipitation areas. In such cases, the features identified by *tobac* in the VIL field well represent individual clouds (i.e., a single detected feature rarely includes more than one cells).

Since VIL integrates reflectivity from the surface to the observed echo top, it better captures hydrometeor condensate in the entire vertical column. This is especially effective for conventional VCP scanning that may miss cells at a specific height if they are very close to the radar or far from the radar. On the other hand, the conventional VCPs that do not include higher elevation angles or that have sparse elevation scans, therefore, tend to produce an underestimation of VIL. Moreover, averaging inhomogeneities within large range-bin volumes, which occur at distances far from the radar, can also cause uncertainties when using VIL. To assess these uncertainties, we investigate the VIL as a function of distance from the radar.

Figure 3 compares contoured frequency by distance distributions of VIL from the 5-min VCP and Full scan (from 0° to 90° over elevation) strategies. Although we use the horizontal distance from the radar instead of altitude in constructing our contoured frequency by altitude diagram, we use the term 'CFAD' to refer to this kind of distribution diagram in this study. Overall, both scans produce small differences in the frequency of less than 0.05 in the CFADs, except within the 30 km range from the radar. For 5-min VCP, there is a shift to higher frequencies of smaller VIL values (red color at distance < 30 km and < -12 dB in Fig. 3b). At distances within 30 km of the radar, both radars have sufficient sensitivity (< -9 dBZ). This underestimation is, therefore, likely due to the fact that 5-min VCP does not observe the upper parts of the clouds. The smaller differences that occur at distances > 90 km, which are shown in both scan strategies, are likely due to the minimum detectable reflectivity, which increases with distance from the radar. It can be concluded that even the NEXRAD VCP

captures the VIL well except for distances less than 30 km from the radar and is, thus, very valuable for the surveillance of
convective cells and is also useful to detect and subsequently track targeted cells, as well as guide the cell tracking using
RHI measurements.

### 3.2  Evolution of polarimetric variables associated with microphysics

Polarimetric observables (e.g., differential reflectivity $Z_{DR}$ and differential propagation phase $K_{DP}$) have frequently been
used by past studies as an indicator of microphysical and updraft evolution (e.g., Kumjian and Ryzhkov, 2008; Kumjian et
al., 2014; Snyder et al., 2013). The NEXRAD polarimetric measurements are very important for capturing the precipitation
microphysical properties. However, its poor spatiotemporal sampling (i.e., limited PPI elevation angles, time for volume
scan) provides only a limited view in convective storms (Fridlind et al., 2019). Here, we assess the impact of the NEXRAD
spatiotemporal sampling by simulating the polarimetric observables from the 1-minute RHI tracking (1-min RHI in Table
1) and the 5-minute conventional PPI volume scan (5-min VCP in Table 1). We randomly select 12 cells from the 453 deep
convective cells tracked in the CLN simulation. These cells all have maximum radar reflectivity exceeding 45 dBZ and 20-
dBZ echo top heights greater than 8 km AGL during their lifetime. We then examine the evolution of microphysical and
dynamical characteristics such as number concentration and mean diameter for each simulated hydrometeor species, as well
as the vertical velocity. Nine of the cells have 40-dBZ mean echo top heights that exceed the freezing level (approximately
5 km AGL) and attain 8 km altitude, which signify stronger convection. These 9 cells show similar evolution of $K_{DP}$, $Z_{DR}$,
and maximum updrafts, all of which have magnitudes greater than 20 m s$^{-1}$ in the middle of their lifetimes. Three of the
twelve cells do not have 40-dBZ echo top heights extending above the freezing level. From the 9 vigorous, deep convective
cells, one representative cell is chosen for a detailed OSSE analysis based on its isolated nature and development near the
NEXRAD radar and other radar locations, used for TRACER and ESCAPE. While we focus on one cell only, the results
can be extended to the other deep isolated cells. Figure 4 shows the evolution of the mass-weighted mean diameter ($D_m$)
and number density for the rain and hail species for the chosen cell. Large rain $D_m$ (> 1.5 mm) is evident near the freezing
level (dashed line) during the later stage of the cell lifetime as the echo top height descends (after 21:50 UTC in Fig. 4c).
Around this time, the largest $D_m$ for hail is also apparent (Fig. 4d). This indicates that the large hail melts as it falls through
the freezing level, thereby producing large raindrops. The hail number concentration (Fig. 4f) is also strongly correlated
with updraft magnitude (Fig. 4b), thus, demonstrating the strong link between the updraft dynamics and hail formation.
Furthermore, the total hydrometeor mixing ratio (Fig. 4a) is consistent with the number concentrations from both rain and
hail (Figs. 4e and 4f).
Figures 5a,d,g (left column in Fig. 5) show simulated reflectivity, $Z_{DR}$, and $K_{DP}$, respectively, averaged over the region
with reflectivity > 40 dBZ from the original, cartesian model grid. The evolution of raindrops as represented by rain $D_m$
(Fig. 4c) is evident by the large values in the $Z_{DR}$ field (Fig. 5d). The relatively large $K_{DP}$ and reflectivity values also seem
to accurately represent the high number concentration of raindrops in the early stage of the cell lifetime (Figs. 4e and 5a,g).
These characteristics of reflectivity, $Z_{DR}$, and $K_{DP}$ are compared with those from the different scan strategies: 1-min RHI
(middle column) and 5-min VCP (right column). The RHI tracking reconstructs the magnitudes and evolution of the
polarimetric observables well (Figs. 5e and 5h) so that they represent the hail $D_m$ and cell evolution (Figs. 4a,b,d).
Meanwhile, the conventional volume scan cannot capture the fine-scale structure and magnitudes of the hail-rain evolution
observed by $Z_{DR}$ and $K_{DP}$ (Figs. 5f and 5i) due to the coarse time resolution. The RHI tracking performs well in capturing
the $K_{DP}$ enhancement and its streak as the raindrops fall (Fig. 5h). Note that the NEXRAD S-band frequency (3.0 GHz) is
assumed for the 5-min VCP simulation, while C-band frequency (5.5 GHz) is assumed for the model and RHI simulation.
Therefore, the $K_{DP}$ values in this figure do include the frequency dependency. The S-band $K_{DP}$ (Fig. 5i) is approximately
1.8 (5.5 GHz/3.0 GHz) times smaller than the C-band $K_{DP}$ (Fig. 5h). This indicates that the $K_{DP}$ measurements from the
shorter-wavelength radar are more sensitive to the $K_{DP}$ evolution, and therefore, can provide more insights on the
microphysical evolution of precipitation.
The region of relatively large $Z_{DR} > 1$ dB extends to 6 km altitude, which is approximately 1 km above the environmental
0ºC level (horizontal dashed line) at around 21:38 UTC (Figs. 5d and 5e). This seems to correspond to the so-called $Z_{DR}$
column (e.g., Kumjian et al., 2014). The $Z_{DR}$ column signature shows more columnar structure in the vertical cross section
at 21:38 UTC (not shown). The $Z_{DR}$ extension is clearly evident in the original model simulation (truth) (Fig. 5d) and the
RHI tracking (Fig. 5e), but it is not clear or is weak in 5-min VCP (Fig. 5f). The large $Z_{DR}$ values associated with raindrops
can be masked by the presence of hail. Hail particles are assumed to be dry and more near-spherical than raindrops following
Ryzhkov et al. (2011) in CR-SIM and dominate the total reflectivity, producing smaller $Z_{DR}$. The $Z_{DR}$ extension is collocated
with large $K_{DP} > 1.8 °$ km$^{-1}$ shown in the original model simulation truth (Fig. 5g) and the RHI tracking (Fig. 5h).

## 3.3 Dynamical evolution

One of the benefits of cell tracking using VIL is that it can better capture the dynamical evolution of convective cells
over their lifetimes (Fig. 2). Figure 6 represents the maximum updrafts in the CLN and POL individual tracked cells as a
function of their lifetime for deep convective cells with 20 dBZ echo top heights exceeding the environmental 0ºC level.
Many of the cells attain maximum updrafts $> 10$ m s$^{-1}$ within the first third of their lifetimes in both the CLN and POL
simulations. The peak occurrence for the POL simulation is found for updrafts that are approximately 5 m s$^{-1}$ stronger than
those of the CLN simulation, suggesting that stronger updrafts are more frequent in the POL than CLN convective cells in
the earlier stages of the cells' lifecycles. Since the earlier stages of convection are driven by warm-phase processes, this
finding is consistent with Marinescu et al. (2021), who found stronger updrafts in the warm-phase region of deep convective
updrafts, but not in the cold-phase region (i.e., above the freezing level) in the POL environment. The stronger updrafts
support the development of larger hail produced in the POL simulation (not shown). This result suggests that it is important
to estimate vertical velocity with a high level of accuracy if the impact of aerosols on convective dynamics is to be properly
resolved in observations. We use the CLN simulation output as well as the individual CLN case deep convective cell shown
in Figs. 4 and 5 to further investigate the uncertainties associated with the multi-Doppler radar vertical velocity retrievals
in this section. Figure 7 shows the maximum updraft velocity in the cell column at each time as a function of the normalized
lifetime for the nine deep convective cells from the CLN simulation selected in the previous section. They all have peak
updrafts exceeding 20 m s$^{-1}$, which mostly occur in the first half of the cells' lifetimes. The black line represents the profile
from the target cell analyzed for the OSSE in this section. It is clear from Figure 7 that the selected cell has a relatively
typical dynamical evolution when compared with the other nine cells, although it does reach its maximum updraft velocity
a little earlier in its lifecycle.
Figure 8 shows the impacts of sets of radar scan strategies for multi-Doppler updraft retrievals for the selected convective
cell using a 3DVAR technique (North et al., 2017; Oue et al., 2019). This cell is the same cell examined in the previous
section (Figs. 4 and 5). We simulate different combinations of the scan strategies using 1-min RHI that scans around the
center of the cell and 5-min VCP. Recall, Table 1 provides the details of the scan strategies, and Figure 1 shows the locations
of the radars with these scan strategies and the targeted OSSE cell. The sets of radars for the multi-Doppler wind retrieval
simulations are: 1) two radars, each using a 1-min RHI (red dot and cross in Fig. 1, called 2RHI); 2) two radars, each using
a 5-min VCP (called 2VCP); 3) two radars, with one using a 1-min RHI (red dot in Fig. 1) and the other using a 5-min VCP
(red cross in Fig. 1) (called RHIVCP); and 4) three radars, with two using 1-min RHIs (red and blue dots in Fig. 1) and one
using a 5-min VCP (red cross in Fig. 1) (called 2RHIVCP). Table 2 represents the root mean square errors (RMSEs) of the
retrieved vertical velocity at four different heights, as well as at all heights. The 2VCP simulation (Figure 8c; green in Figure
8f) significantly underestimates the updraft, with the error exceeding 5 m s$^{-1}$ above 5 km AGL, where the cell produces
mean updrafts stronger than 12 m s$^{-1}$. The 2VCP radar pair, whose volume scan takes 5 minutes, does not resolve the updraft
evolution well. We note that other studies also found an underestimation of vertical velocity retrievals using two 5-min
VCPs. For example, Marinescu et al. (2020) used two 5-min VCPs to estimate strong updrafts in supercells and found an
underestimation in the region from 5-10 km AGL when compared with radiosonde estimates of vertical velocity. This pair
of 5-min VCPs (2VCP) does, however, produce less error below 4 km AGL where the cell produces weaker updrafts (< 5
m s$^{-1}$) when compared with the other sets of radar combinations. This suggests that the conventional PPI scans, which have
dense scans at low elevation angles, well capture the low-level horizontal inflow, and the mass continuity assumption is
well satisfied at the low levels. It is interesting that while 5-min VCP represents VIL well for the distance > 30 km as shown
in Fig. 3, its limitations produce significant uncertainties in the convective dynamical retrieval of individual clouds above
~5 km AGL even though the cell is observed at a distance > 30 km from the radar (Fig. 1).
With an RHI scan every minute, even when adding only one RHI, cell tracking improves the retrievals above 5 km altitude
(Figs. 8b,d,e; 2RHI, RHIVCP, and 2RHIVCP; red, magenta, and blue, respectively in Figure 8f). The improvements are
particularly significant for regions in which the updraft velocities are stronger than 10 m s$^{-1}$. The RHIVCP simulation shows
the best estimate at the middle altitude (~6 km) among the four simulations, followed by 2RHIVCP, and thirdly 2RHI. The
2RHI and 2RHIVCP simulations show RMSEs less than 6 m s$^{-1}$ at all altitudes and better estimates than the other two
simulations at the higher altitudes (8 and 10 km AGL). The RHI scan has better sampling in the higher elevations than 5-
min VCP, resulting in a better retrieval at these higher altitudes.
As the profile and Table 2 show, 2RHI and 2RHIVCP have the lowest RMSEs when considering all altitudes (Table 2,
bottom row). In addition, 2RHIVCP shows better results at altitudes < 10 km than 2RHI. This suggests that the conventional
5-min VCP scan can be used for further improvement of the RHI-only tracking retrievals for the low and middle altitudes.
Since the 5-min VCP has dense scans at lower elevations, this can help to provide enough data covering the horizontal
domain of the cell, which may better represent the low-level horizontal wind convergence, thereby, better constraining the
cost functions in the 3DVAR.
We also investigate the impacts of the radar radial locations relative to the same cell as in Figs. 4, 5 and 8. Radars
horizontally extending from 10 to 70 km (in 10 km increments) radially away from the cell are assessed. For this analysis,
we use the scan strategy with the lowest errors from our prior analysis, i.e., two radars performing 2-min SECs and one
radar performing 5-min VCP (e.g., Table 2, the tracking radars used 2-min SEC rather than 1-min RHI, We believe that 1-
min RHI can be feasible with electrical scan or mechanical rapid scan radars. However, 2-min SEC can be more reasonable
when the cell is relatively close to the radars, because scans need to extend to higher elevations as discussed in Sect. 2.4.2).
Figure 9a shows the radar locations for the seven simulations and Figure 9b demonstrates the vertical profiles of errors of
the retrieved updrafts averaged over a 20 km x 20 km box with reflectivity > 30 dBZ at 21:42 UTC. For each retrieval, the
largest error is evident above an altitude of ~9 km AGL where the stronger updrafts are simulated by the model (Fig. 8a).
The largest error among the retrievals is found in the retrieval with the radars closest to the cell (red profile in Fig. 9b). This
occurs since the PPI volume scan does not cover the upper part of the cell and/or the horizontal wind convergence at higher
elevation angles may not be retrieved from the RHI measurements accurately. When each radar has a distance greater than
or equal to 20 km from the cell, the retrievals are improved by 5-10 m s$^{-1}$ between 5 to 11 km altitudes. The retrievals in
which the radar distances from the cell fall between 20 and 50 km show errors less than 5 m s$^{-1}$ below 11 km AGL. Such
accuracies in the retrievals may allow for resolving the aerosol impacts on updraft velocities shown in Fig. 6. The errors are
then found to increase again above 10 km AGL, especially for the radars located 60 and 70 km away from the cell. This
investigation suggests that the radars should target cells that are between 20-50 km from the radar for optimal multi-Doppler
radar retrievals. This finding is consistent with previous field campaigns using multi-Doppler radar measurements (e.g.,
Wurman et al., 2012; Collis et al., 2013; Jensen et al., 2016) and OSSE studies (e.g., Potvin et al., 2012a).
In the simulations above, the three radar locations are almost equidistant from the target cell. Now we explore the
impacts of having radars located at different distances from the target. We move one of the three radars to a distance of 10
to 70 km at 10 km increments (except 20 km which has already been tested) while keeping the other two radars as a fixed
distance of 20 km (blue dots in Fig. 9a). Similar to Fig. 9b, Figs. 10a and 10b show vertical profiles of the errors of the
retrieved updrafts when moving the 2-min SEC radar (at the northwest corner of the triangle) and the 5-min VCP radar (at
the south corner), respectively. When moving the 2-min SEC radar from distances between 30 and 70 km (Fig. 10a), the
retrievals show better profiles as the RMSEs range from 1.7 to 2.6 m s$^{-1}$. The RMSE increases when the radar is located at
10 km, which is consistent with the equidistance simulations (Fig. 9). Another notable point is that when the radar is located
at distances from 50 to 70 km, the errors below 1 km slightly increase, which is most likely because the radar coverage is
sparse at the lowest elevation due to the distance. Similarly, when moving the 5-min VCP radar, the RMSE increases when
the radar is located at 10 km (Fig. 10b). The impact is significant above 5 km altitude. When the radar is located at 60 or 70
km the errors below 5 km increase. This also reflects the sparse radar coverage at the lower altitudes for the far distances.

In nature, convective cells often do not nicely evolve over pre-defined multi-Doppler regions and move outside the

region of optimal analysis. Therefore, we also propose a single-RHI vertical velocity retrieval which can be used on a much
larger sample of convective cells in the vertical in the vicinity of the radar compared to fixed, multi-Doppler platforms. The
single-RHI vertical velocity retrieval extracts the vertical air motion component from the radial velocity (Doppler velocity)
which is composed of the vertical air motion, horizontal air velocity, and hydrometeor fall velocity (Lamer et al., 2014).
Figure 11a shows examples of Doppler velocity vectors (clear arrows) and the components of the Doppler velocity including
horizontal wind along the RHI plane (yellow arrows), vertical velocity (red arrows), and hydrometeor fall velocity (blue
arrows) at two different points ([x,y]=[0 km, 5 km] and [7.5 km, 5 km]). These examples assume that each component at
the two points has the same value. At the radar distance equal to 0 km (x=0 km), the horizontal wind component can be
ignored. At the radar distance greater than 0 km, the contribution of the horizontal wind component increases with
decreasing elevation angle (i.e., increasing the distance from the radar at a constant height). To apply this technique to real
observations, horizontal velocity and hydrometeor fall velocity should be provided. Generally, the horizontal velocity profile
can be provided from a velocity-azimuth display (VAD) technique using PPI measurements or sounding measurements,
assuming that the horizontal wind is constant at each level. However, this assumption is a major source of the uncertainty
in the single-RHI vertical velocity retrieval technique, particularly at lower elevation angles. At these lower elevation angles,
the horizontal wind component dominates the radial velocity, but the coverage of these lower elevation angles often does
not properly capture the variability in the horizontal wind, especially close to the radar. We, therefore, investigate the impact
of the distance of the radar from the cell on the single-RHI retrieval. In the simulations, we use the reflectivity-weighted
hydrometeor fall velocity simulated by CR-SIM, similar to the present multi-Doppler retrieval simulations, to exclude the
uncertainty related to the fall velocity estimates.

Figure 11 shows the simulated single RHI vertical velocity retrieval from the selected convective cell. Profiles in Figs.

11c and 11d are retrieved vertical velocity at the convective core (distance = 0 km) and the errors from the truth, respectively.
We investigate this technique for a profile at 21:42 UTC of the cell (same as Figs. 8f and 9b), where the strongest updraft
is simulated. This single-RHI Doppler velocity technique works very well at the distance = 0 km (red), where the horizontal
wind component can be ignored, as evidenced by the error profile being equal to 0 at all altitudes (red line). However, below
6 km AGL, the error significantly increases with the radar distance from the core. Interestingly, the characteristics of the
error distribution are opposite to those of the multi-Doppler retrievals (Figs. 8f and 9b). We would, therefore, suggest the
complementary use of the multi-Doppler wind retrieval and the single-RHI vertical velocity retrieval for better vertical
velocity estimates of convective cells. For example, in a tracking strategy in which two radars track a targeted cell, the
optimal scenario can be one in which the two radars track the cell with sector RHI/PPI scans at intervals of ~2 min when
the distance of the cell from both radars is greater than 20 km. However, when the distance of the cell from one of the radars
is less than 20 km, the radar's scan is then switched to hemispheric RHI.
This study highlights the importance of focusing on high-spatiotemporal observations of individual convective cells
rather than utilizing conventional surveillance scans. Such high-spatiotemporal observations can be accomplished by
tracking cells using fast scan RHI measurements facilitated by rapid-scan radars. However, it is not hard to anticipate that
the number of individual cells tracked successfully during a short-term intensive observation period where such special scan
strategies are performed will also be limited. Therefore, we have investigated the sample size of cells needed to represent
the typical convective evolution of deep convective cells using the median maximum updraft metric shown in Fig. 6. This
specific analysis accounts for the error regarding cell sampling, but it does not account for the wind retrieval uncertainty
from the scan strategy. Figures 12a and 12c show boxplots of the maximum vertical velocity as a function of the normalized
lifetime from all convective cells detected (910 tracked cells including deep and shallow cells) in the CLN case and from
deep convective cells defined in Fig. 6a (453 tracked cells), respectively. These figures indicate high variability in the
maximum updraft magnitude as a function of time, and that potentially, one randomly sampled convective cell may not
represent the typical evolution of vertical velocity. Figure 12b depicts the relationship between the sample size and the
errors associated with estimating the full population median evolution of the maximum updraft magnitude. We randomly
sample convective cells from all of the *tobac*-detected cells in the CLN simulation (910 tracked cells) and estimate the
median value of maximum updrafts at each time bin with different numbers of samples. The median values for the different
sample sizes are then compared to the median values from all deep convective cells detected (shown as a black line in Fig.
6a) to estimate RMSEs. Figure 12b suggests that increasing the sample size generally decreases the RMSE to less than 4.5
m s$^{-1}$ until a population of 10 cell samples is reached and converges to approximately 2.6 m s$^{-1}$ for a sample size of 20 or
more samples. When focusing the analysis on deep convective cells only (Fig. 12d), the RMSE decreases to approximately
3 m s$^{-1}$ for 10 cell samples and converges to approximately 1 m s$^{-1}$ for 40 or more samples.
This study focuses on tracking isolated deep convective cells, each of which has a single core. Although we provide a
detailed investigation of one selected cell using OSSEs, the result should be robust for the other cells that have a similar
vertical structure to that shown in Sect. 3.2. The error values presented in this study, however, may depend on cloud type.
As the larger errors of the multi-Doppler radar wind retrievals are shown to exist in the higher altitudes in this study, the
heights of convection could influence the observational uncertainties (i.e., height of maximum updraft). Moreover, in a
strong wind shear environment where storms advect quickly, the impact of the use of quick updates of RHI scans would be
more effective (e.g., Clark et al., 1980; Oue et al., 2019). Various convective cloud morphologies have been investigated in
terms of uncertainties in observations, including mesoscale convective systems (e.g., Bousquet et al., 2008; Oue et al.,
2019), supercells (e.g., Potvin et al., 2012; Marinescu et al., 2020), and convection embedded in stratiform precipitation
(e.g., Bousquet et al., 2008). However, the qualitative characteristics found in this study, such as the error profile trends, the
dependency on the radar locations, and the dependency on scan strategy, are likely to be common to those deep convective
cloud systems, as well.

**4 Summary**



Optimizing radar observation strategies has been one of the most important topics in pre-field-campaign periods,
especially when the focus is on atmospheric phenomena that rapidly evolve on timescales that standard operational radar
networks cannot resolve. This study uses the Cloud-resolving Radar Simulator (CR-SIM) and the *tobac* cloud object
tracking algorithm to investigate observational uncertainties of isolated deep convective clouds associated with pre-existing
and planned radar deployments and strategies. The focus of this manuscript is to optimize the radar observation strategies
for the TRACER/ESCAPE field campaign, but the results are also generalizable for field campaigns that focus on isolated
deep convection using radar observations.
The following results and associated recommendations are made:
● The cell tracking algorithm with the use of VIL better captures the difference in cell lifetimes between the low-CCN
(CLN) and high-CCN (POL) simulations compared with the use of reflectivity thresholds at individual altitudes and
is suitable to detect and track more convective cells for longer time periods, including the early-developing and
dissipating stages of isolated storms.
● An analysis of the CLN and POL simulations, used to quantify the impact of aerosols on the convective dynamical
evolution, show a 5-7 m s$^{-1}$ difference in maximum updraft at the early stages of convective development. This
suggests the importance of accurate vertical velocity estimates using the radar observations if the impact of aerosols
on convective updrafts is to be assessed.
● Fast scanning of the individual convective cells every minute captures the microphysical evolution better than the
operational radar observations that update the volume scan every 5 min. In particular, the tracking of cells using RHI
every minute better captures the evolution of $K_{DP}$ in the early stage and $Z_{DR}$ in the later stage, which are primarily
associated with the rain number concentration and hydrometeor particle (hail and rain) size, respectively.
● Tracking using RHI improves the multi-Doppler radar updraft retrievals above 5 km AGL, particularly for regions
with updraft velocities greater than 10 m s$^{-1}$. The conventional 5-min PPI volume scan can be used for further
improvement of the RHI-tracking-only retrievals.
● The multi-Doppler radar updraft retrievals, even when using RHI, are still challenging, especially for cells that are
close to the radars (i.e., within 10 km of the radar). This approach can be complemented by a single RHI updraft
retrieval technique.
● Based on these results, the suggested strategy to better capture microphysics and dynamics of deep convective cells
is tracking by frequent RHI scans from more than one radar (blue and red scans in Fig. 13), in addition to the
operational PPI volume scans generally performed by the NEXRAD radars (green scans in Fig. 13). We also suggest
a hybrid radar scan strategy which switches between the RHI cell tracking and hemispheric RHI measurements
depending on the distance between the radar and the targeted cell (red and orange scans in Fig. 13). Such RHI tracking
measurements would be possible with conventional mobile radars, but the fast-scanning Doppler radars (Wurman,
2001), and/or phased array radars (Kollias et al., 2022) would have more advantages in faster updating, better spatial
resolution, and higher quality datasets.
● Increasing the number of deep convective cells sampled by such observations better represents the population median
maximum updraft evolution. When increasing the number of deep cells sampled to more than 10, the RMSE
decreases to less than 3 m s$^{-1}$, and when increasing the sample size to more than 40, the RMSE further decreases to
less than 1 m s$^{-1}$.

For the strategy suggested above we have assumed that the real-time cell tracking will be guided by another algorithm
that will take advantage of surveillance scans by conventional radar networks (e.g., a Multi Sensor Agile Adaptive Sampling
(MAAS) framework (Kollias et al., 2020). The new MAAS has incorporated a cell-tracking algorithm using a watershed
technique (similar to tobac and the approach of Hu et al., 2019) and predicts the future location of convective cells using
multiple sensors (e.g., NEXRAD radar at Houston, TX and GOES-16).
Finally, this study highlights the importance of using OSSEs in developing radar strategies during pre-field campaign
periods. Current radar systems used for field campaigns, as well as operational radars, have more functions (e.g.,
polarimetry, Doppler, Doppler spectrum, and dual wavelength) and configurable parameters (e.g., pulse sampling, pulse
width, range-bin gate, azimuth, elevation spacings) than in the past. While this increased functionality makes the scan
strategies more sophisticated, it also makes the optimization of the scan strategy more complex. Although we argue that the
results from this study can be applied to other field campaigns that focus on deep convection, for more qualitative analyses,
the pre-field campaign OSSEs should also be optimized for a specific field campaign thereby accounting for characteristics
of the radar systems that will be used for the field campaigns. The use of a radar simulator in the OSSEs provides several
advantages including 1) facilitating instrument deployments, such as the radar locations and the number of radars required,
and accounting for the radar characteristics and functions; 2) optimizing radar configurations such as the scan rate, elevation
angles, update time of scans, and trade-offs; and 3) quantifying errors of the observables and retrievals. Effective OSSEs
can lead to successful, state-of-the-art field campaigns and provide high-quality, unique datasets that can allow for new
insights of the atmospheric phenomena.


*Code availability*. The source code and user manual for the Cloud Resolving Model Radar Simulator (CR-SIM) are available
at https://www.bnl.gov/CMAS/cr-sim.php, last access: 21 April 2022, and those for Tracking and Object-Based Analysis
of Clouds (*tobac*) are available at https://tobac.readthedocs.io/en/latest/.

*Data availability*. The ACPC model intercomparison project deep convection simulation data used for the input of CR-SIM are stored and can be accessed on the U.K. CEDA JASMIN supercomputer. Vertically integrated liquid (VIL) products from CR-SIM used for the convective cell tracking are available in Stony Brook University Academic Commons (https://commons.library.stonybrook.edu/somasdata/16, last access July 29, 2022).

*Author contributions*. The radar simulator and cell tracking work and analysis were made by MO. Conceptualization of the method, interpretation, and writing were shared between MO, PK, SMS, PJM, and SCV. The radar simulator was developed by MO and PK's group, and the cell tracking code was developed by SCV's group.

*Competing interests*. The authors declare that they have no conflict of interest.

*Acknowledgements.*
M. Oue, S. M. Saleeby, and S. C. van den Heever were supported by Atmospheric System Research (grant no. DE-SC0021160). M. Oue and P Kollias were also supported by National Science Foundation Grant FAIN-2019932.

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

Table 1. Radar scan strategies simulated in this study.

| Strategy | Full elevation scan for an azimuth sector tracking cells (1-min RHI, 2-min SEC) | 5-min volume coverage pattern (5-min VCP) | Full elevation/azimuth scan (Full) |
|---|---|---|---|
| Beam width | 1.0° | 0.9° | 1.0° |
| Elevation angles | From 0.5° to 89.5° every 1° | 0.48, 0.88, 1.32, 1.8, 2.42, 3.12, 4.0, 5.1, 6.42, 8.0, 10.02, 12.48, 15.6, and 19.51° | From 0° to 90° every 1° |
| Azimuth range | 14.5° at 40 km radar range (Sector to cover a 10-km width centered around the individual cells with 1° spacing) | From 0° to 360° with a 0.5° increment | From 0° to 360° with a 1.0° increment |
| Time for volume scan | 1 minute or 2 minutes* | 5 minutes | 1 minute |

*With the radar beam width of 1°, the total number of beams for the sector scan is 90 (over elevation) x 14 (over azimuth)
= 1260 beams. Assuming that each beam needs ~96 radar samples, the total number of pulses is 120960. This takes 1-2
min with typical pulse repetition ratios (1.5 - 2.5 kHz) for C- and X-band radars. See detailed discussion in Sect. 2.4.2

Table 2. The root-mean-square error (RMSE) of the retrieved updraft averaged over the regions with reflectivity ≥40 dBZ
at four different altitudes as well as all heights for a variety of scan strategies for the entire lifetime.

|  | 1. Two 1-min RHIs (2RHIs) | 2. Two 5-min VCPs (2VCPs) | 3. One 1-min RHI + one 5-min VCP (RHIVCP) | 4. Two 1-min RHIs + one 5-min VCP (2RHIVCP) |
|---|---|---|---|---|
| 10 km | 4.794 | 16.82 | 7.995 | 4.800 |
| 8 km | 5.371 | 7.396 | 5.609 | 5.112 |
| 6 km | 5.862 | 6.601 | 4.764 | 4.895 |
| 4 km | 4.232 | 3.178 | 3.625 | 3.511 |
| All heights | 5.030 | 6.763 | 5.539 | 4.535 |


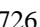


**Fig. 1: (a) A snapshot of the RAMS-simulated total hydrometeor condensate field at 21:09 UTC at 5.5 km ASL; (b) CR-SIM simulated radar reflectivity field at the same height and same time as (a); (c) vertically integrated liquid (VIL) estimated from the CR-SIM C-band total reflectivity (from total liquid and ice hydrometeor condensate) at the same time as (a); and (d) tracks of precipitating convective cells detected between 20:00 and 23:59 UTC using _tobac_. On each panel, the red "X" marks the location of a radar performing 5-min VCP (i.e., NEXRAD KHGX), the red solid dot represents the location of a radar performing a different 5-min VCP or RHI, and the blue solid dot represents the location of another radar performing RHI. The red rectangle represents the tracked cell of interest used for multi-Doppler radar retrieval and polarimetric evolution analysis.**

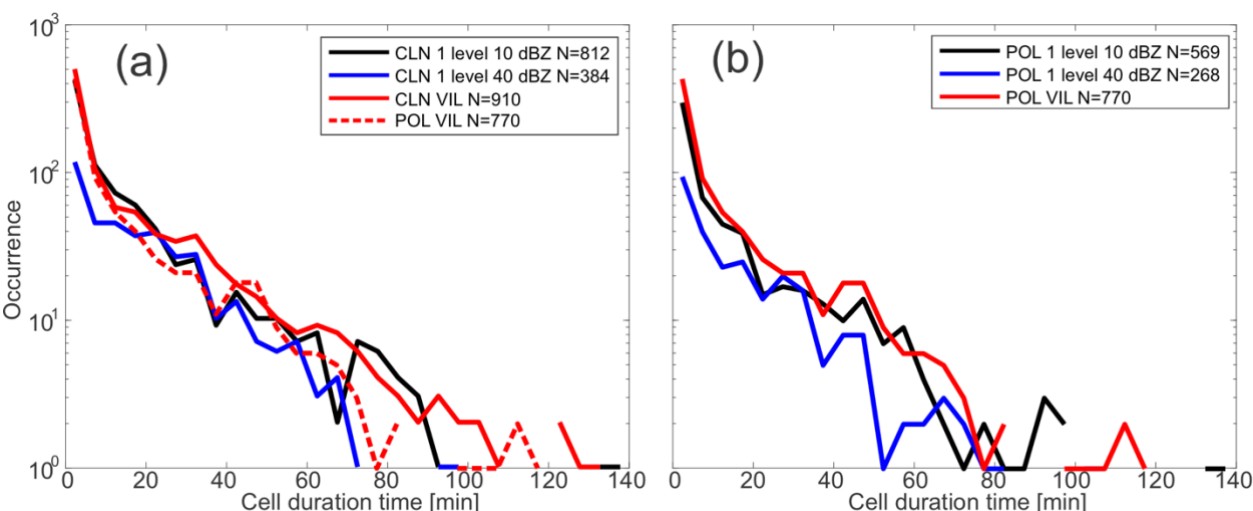

**Fig. 2: Frequency distributions of cell duration time from the _tobac_ cell tracking using VIL (red), 10 dBZ threshold at 2 km height (blue), and 40 dBZ threshold at 2 km height (black) for (a) CLN and (b) POL cases. The legend displays the total number of detected cells (N) for each tracking parameter utilized. Panel (a) also includes the cell tracking using VIL for the POL case shown as the red dashed line. The time bin size for the frequency distribution plots is 5 min.**

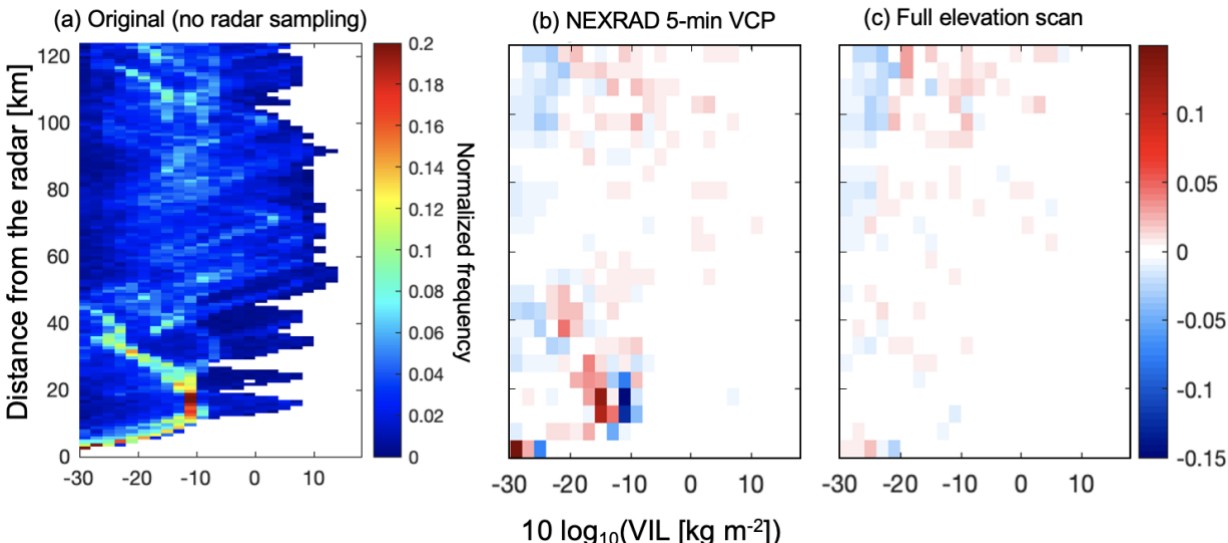


**Figure 3: (a) Contoured frequency by distance (from the radar) distribution of the VIL from the original, cartesian model grid**
**from the 1-minute output over the 4-hour analysis period; (b) difference between the VIL from the 5-min VCP scan strategy and**
**(a); and (c) difference between the VIL from the Full scan strategy and (a). The VILs from the 5-min VCP and Full scan strategies**
**are estimated from the gridded reflectivity fields with 250 m horizontal and vertical spacing and 1-minute output over the 4-hour**
**period.**

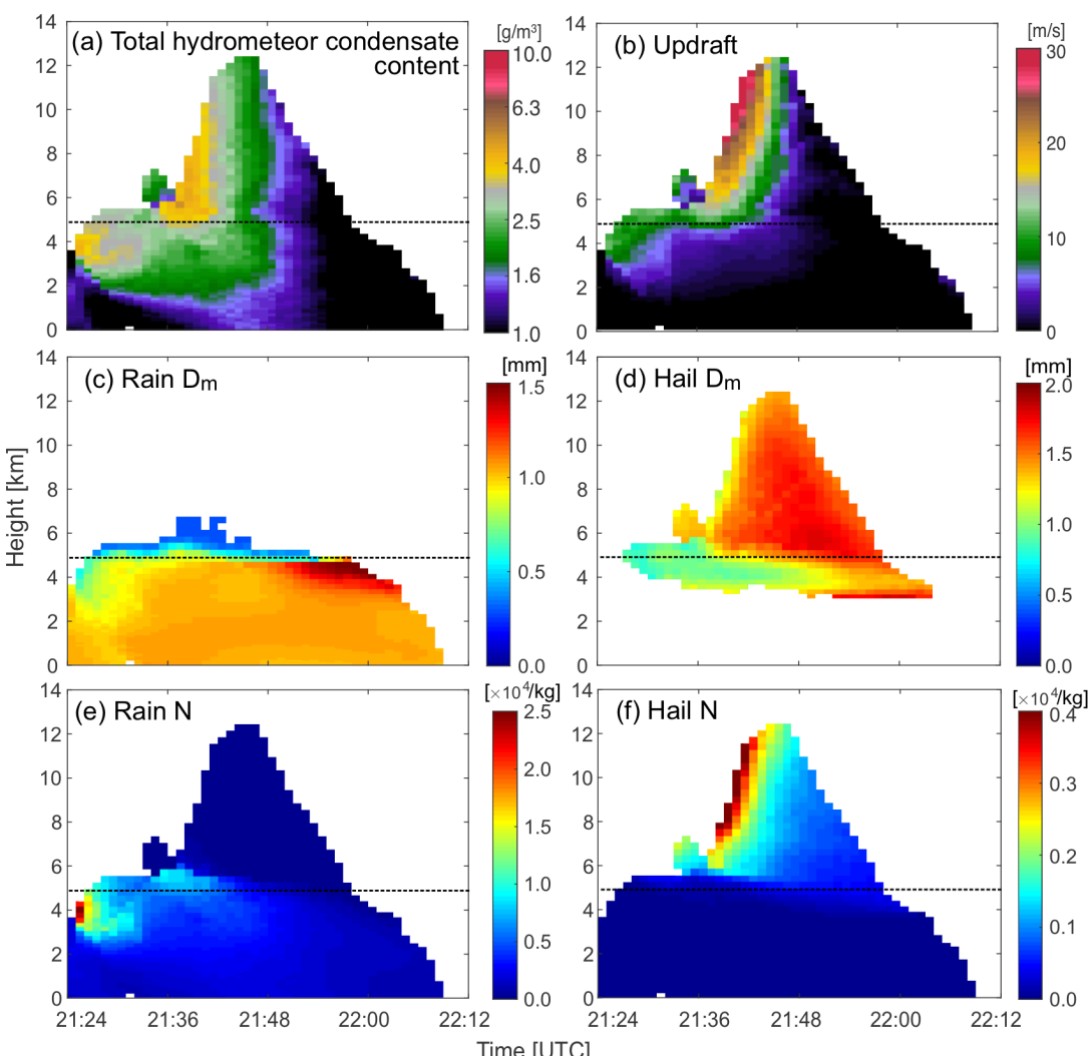

**Fig. 4: Height-versus-time cross sections of the (a) total hydrometeor condensate content, (b) updraft, (c-d) the mass-weighted**
**mean diameter ($D_m$) for (c) rain and (d) hail, and the number density (N) for (e) rain and (f) hail, averaged for areas with**
**reflectivity > 40 dBZ of the selected convective cell from the CLN case. Dashed line in each panel represents a 0°C isotherm**
**of domain-averaged temperature.**

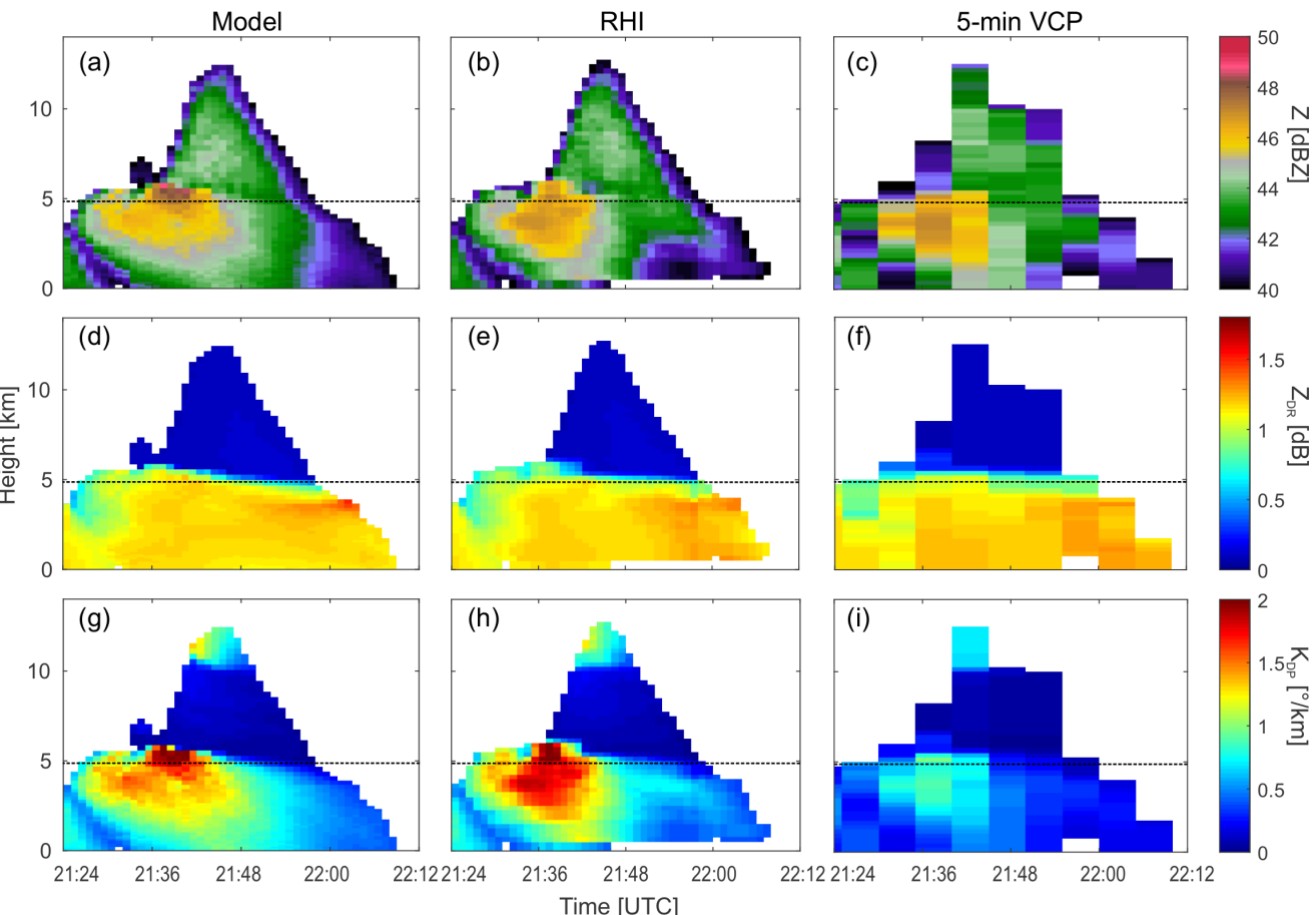


**Fig. 5: Time-height cross sections of C-band radar reflectivity (top row), Z_DR (middle row), and K_DP (bottom row), averaged for**
**areas with reflectivity > 40 dBZ for the selected convective cell for (a,d,g) the model simulation truth, (b,e,h) simulated RHI**
**tracking strategy, and (c,f,i) simulated 5-min volume scan strategy. The cell in this figure is the same as that shown in the box in**
**Fig. 4 and is from the CLN case. Note that the NEXRAD S-band frequency is assumed for the 5-min VCP simulation, while C-**
**band frequency is assumed for the model and RHI simulation. Therefore, the K_DP values in this figure include the frequency**
**dependency. Dashed line in each panel represents a 0°C isotherm of domain-averaged temperature.**

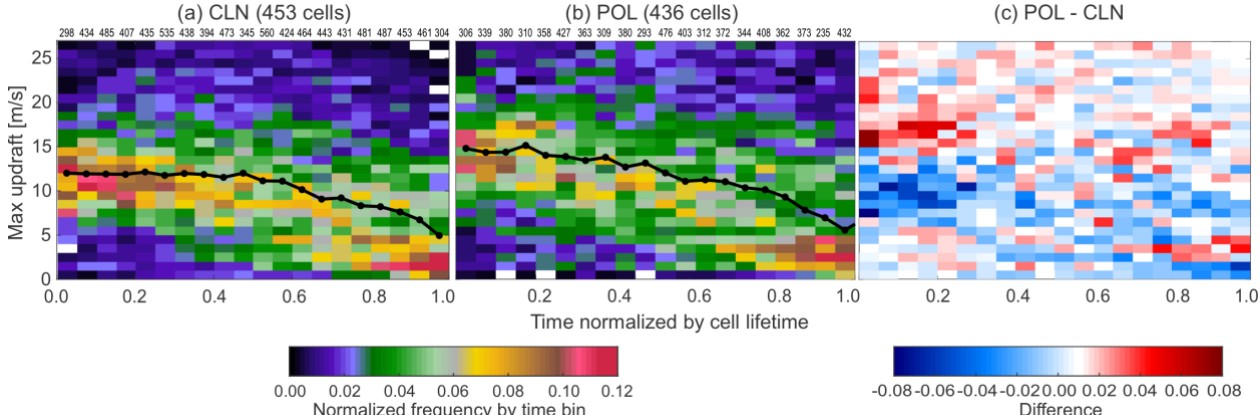


**Fig. 6: Frequency of maximum updraft magnitudes in detected individual cells as a function of time normalized by the cell duration for the (a) CLN and (b) POL case simulations, and (c) the difference between the CLN and POL cases (POL – CLN). Here we present only those deep convective cells with 20 dBZ echo top heights that exceeded the freezing level during their lifetimes. Color shading in (a) and (b) represents normalized frequency by cell lifetime, and that in (c) represents the difference in the normalized frequency. The sample size at each time bin is presented on the top of (a) and (b). Black lines in (a) and (b) represent the median value in each time bin.**

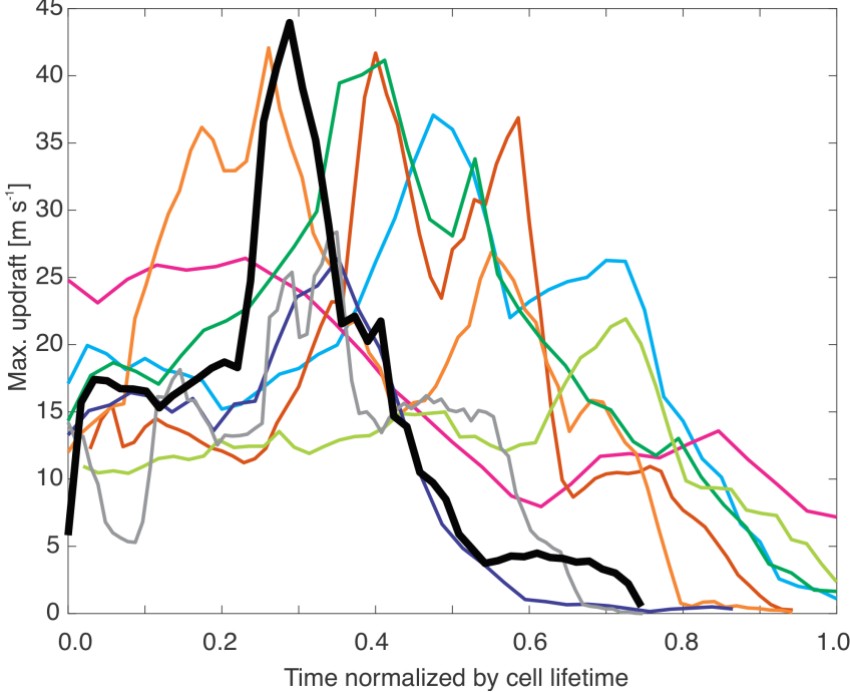

768

**Figure 7: Maximum updraft velocity in the cell column at each time represented as a function of the normalized lifetime for the nine deep convective cells from the CLN simulation. These cells were randomly selected, as described in Section 3.2, and were required to have the maximum radar reflectivity greater than 45 dBZ, the echo top height of 40 dBZ exceeding 5 km in altitude,**

 and the echo top height of 20 dBZ extending above 8 km altitude during the storm lifecycle. The black line represents the target

cell that was analyzed for the present OSSE. Note that because the plot displays the maximum updraft found in regions with
reflectivity greater than 45 dBZ, some lines do not end at time=1.0 when the maximum reflectivity is below 45 dBZ.

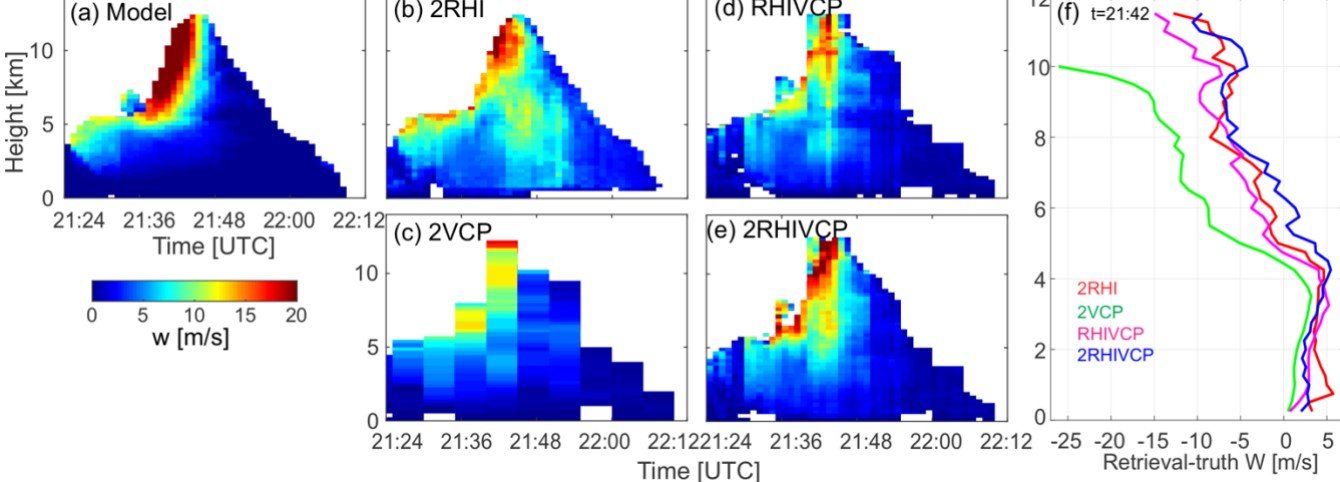


**Fig. 8: Height-time cross sections of the updraft velocity averaged over the area with reflectivity > 40 dBZ from (a) the model**
**(truth) and (b-e) the simulated retrievals, as well as the (f) errors of the simulated multi-Doppler vertical velocity retrievals**
**(retrieval - truth) at 21:42 UTC, when the maximum updraft was produced by the cell.**

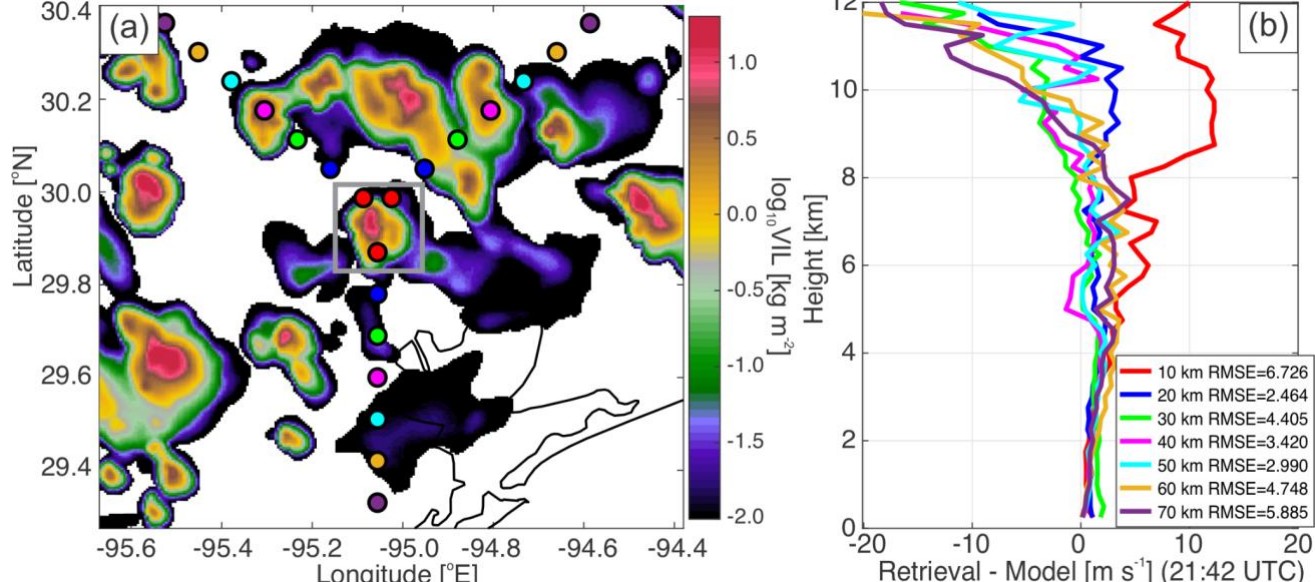



**Fig. 9:** (a) Horizontal distribution of VIL centered around one identified convective cell (gray box, the same cell shown in Figs. 4, 5, and 8) at 21:42 UTC from the CLN simulation and (b) vertical profiles of errors of simulated retrievals (retrieval - model) averaged over a region with reflectivity > 30 dBZ at 21:42 UTC for the identified convective cell. The colored dots in (a) represent the radar locations for the multi-Doppler radar wind retrievals. The colors of the dots correspond to the colors of the set of the radars for the multi-Doppler radar wind retrievals shown in (b). The two radars to the north of the cell performed 2-min RHIs, and the other performed 5-min VCP. The RMSE for each profile is displayed in (b).

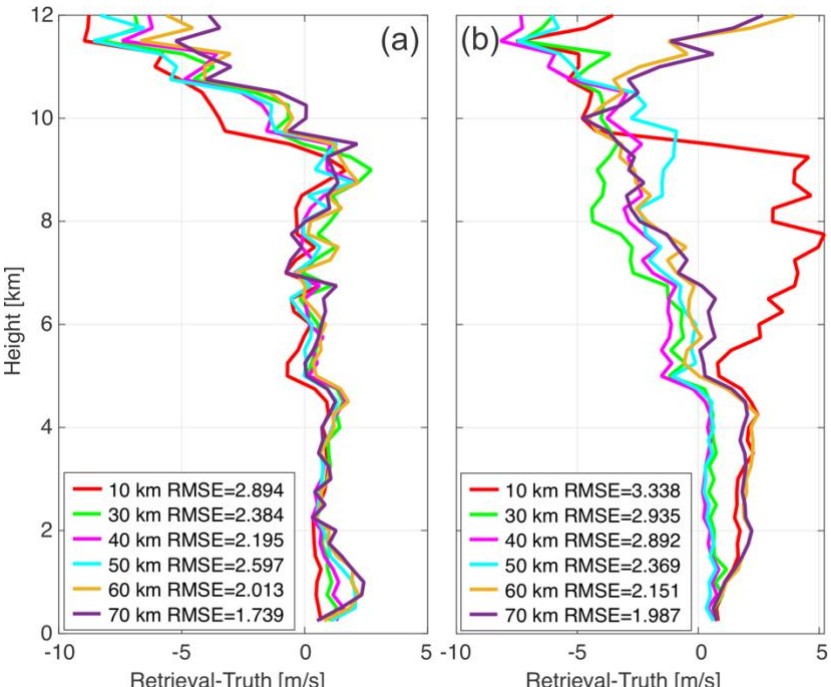

**Fig. 10:** Vertical profiles of errors of simulated retrievals (retrieval - model) averaged over a region with reflectivity > 30 dBZ at 21:42 UTC for the identified convective cell using the three radars where the two are fixed at distance = 20 km (blue dots in Fig. 9a) while one is moved (dots with the other colors in Fig. 9a). In (a) the 2-min SEC radar (at the northwest corner of the triangle in Fig. 9a) is moved, and in (b) the 5-min VCP radar (at the south corner of the triangle in Fig. 9a) is moved. The colors of lines correspond to the dot's colors for the moved radar.

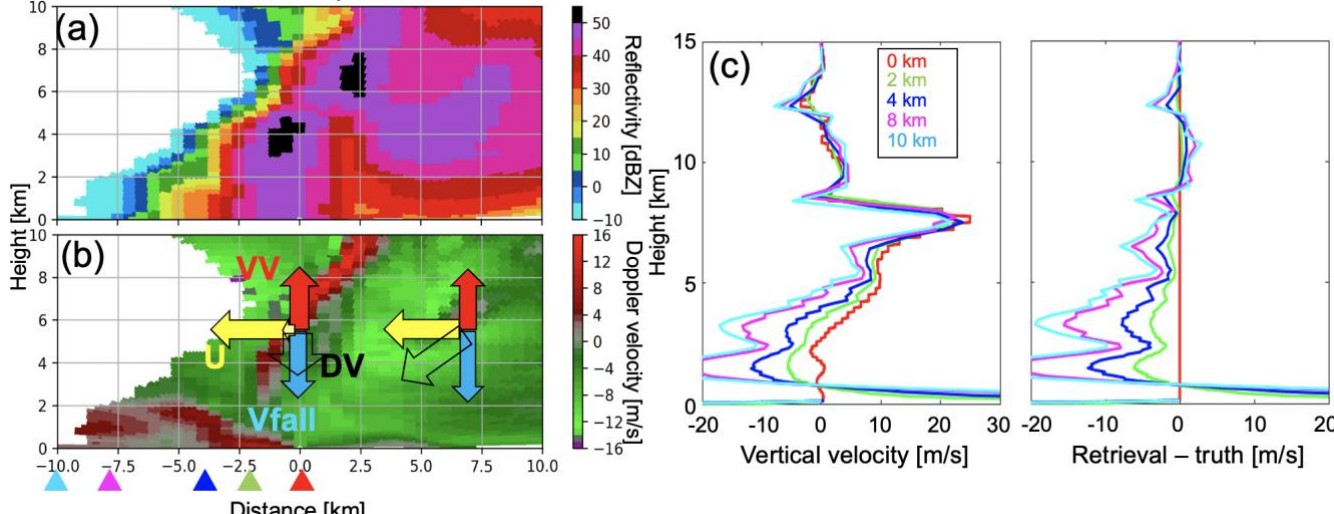

Fig. 11: Vertical cross section of (a) radar reflectivity and (b) Doppler velocity from the simulated RHI measurement for a convective cell and vertical profiles of (c) retrieved vertical air motion and (d) errors (retrieval – model), simulated with different distances between the radar and the center of the convective cell (distance = 0 km in b) at 21:42 UTC. The location of the radars from the center of the convective cell in (c-d) are indicated by their corresponding colored triangle in panel (b). A negative Doppler velocity in (b) represents motion toward the radar. In Panel (b), arrows represent examples of the Doppler velocity vector and the components at two range-height bins ([x,y]=[0 km, 5 km] and [7.5 km, 5 km]); the clear arrows with black line represent the observed Doppler velocity (labeled as DV), the red arrows represent vertical velocity component (labeled as VV), the yellow arrows represent horizontal wind component along the vertical cross section (labeled as U), and the light blue arrows represent the component of the particle fall velocity (labeled as Vfall). These examples assume that each component at the two points has the same value, but the scale does not represent a specific value.

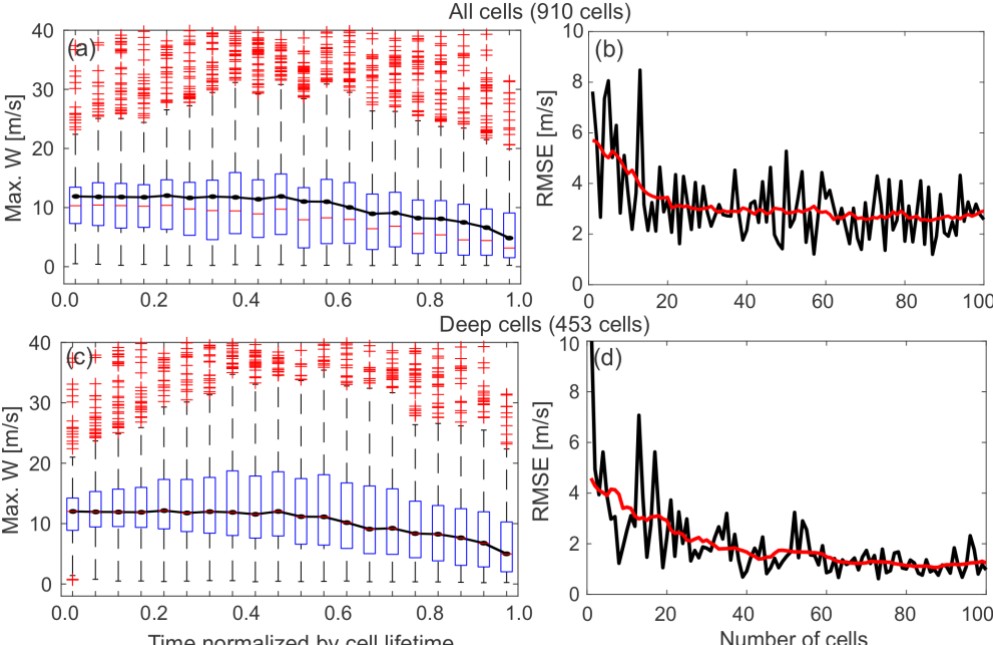

Figure 12: (a,c) Boxplots of maximum vertical velocity as a function of the normalized lifetime from all convective cells detected (910 cells including deep and shallow cells) from the CLN case and from the deep convective cells (453 cells) defined in Fig. 6a, respectively. For each boxplot, the central red mark indicates the median, and the bottom and top edges of the box indicate the 25th and 75th percentiles, respectively. The whiskers extend to the most extreme data points excluding outliers.  Outliers are plotted individually using the cross symbol. Black solid lines in (a) and (c) represent the temporal evolution of the median values of maximum updrafts for deep cells as a function of time. (b,d) The RMSEs of median values of the maximum vertical velocity as a function of the number of cells randomly sampled from all convective cells detected in the CLN simulation (b) and from deep convective cells defined in Fig. 6a (d). The RMSEs are estimated from the median profiles as a function of the normalized lifetime from the random sampling and that from all deep convective cells (black line in Fig. 6a).

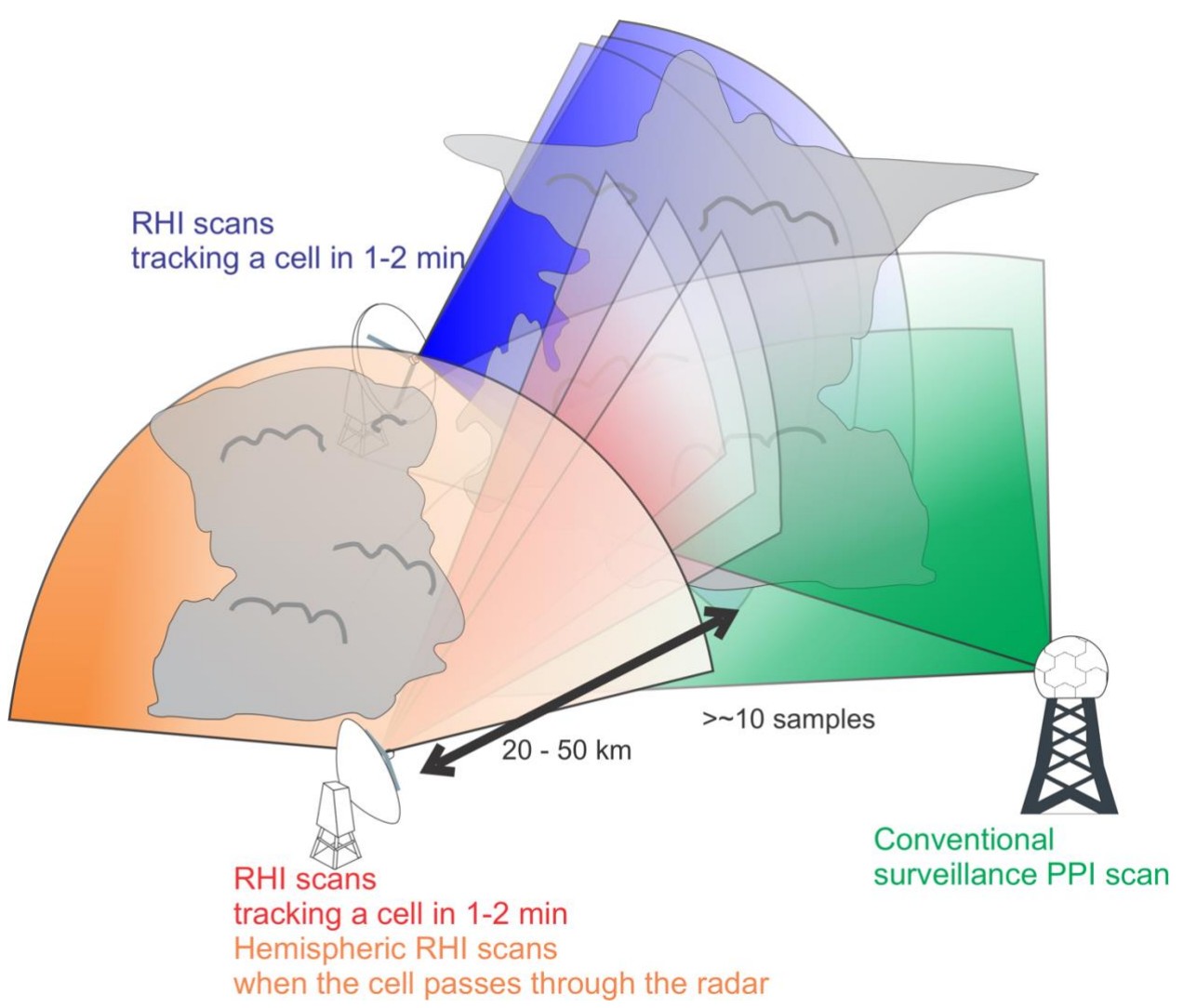

RHI scans
tracking a cell in 1-2 min

>~10 samples

20 - 50 km

RHI scans
tracking a cell in 1-2 min
Hemispheric RHI scans
when the cell passes through the radar

Conventional
surveillance PPI scan


**Figure 13: A schematic image of a suggested scan strategy optimized for observing convective cell evolution. Optimal cell tracking**


**is achieved by frequent RHI scans from more than one radar (blue and red scans) in addition to the operational PPI volume scans**


**generally performed by the NEXRAD radars (green scans). The schematic also suggests an optimal hybrid radar scan strategy**


**which switches between cell tracking by frequent RHI measurements and hemispheric RHI measurements depending on the**


**distance between the radar and the target cell (red and orange scans).**


