# Peer review of "Optimizing Radar Scan Strategies for Tracking Isolated Deep"

_EGUsphere, 2022_

## Author Comment (AC1)

Responses to Referee #1

Our response is highlighted by blue below each Referee comment in black.

The preprint by Oue et al focuses on the choice of radar scanning strategies for campaigns aiming at investigating convection processes. The study is conducted using and Observing System Simulation Experiment referred to a 4-hours event. RHI sweeps are recommended to complement observation provided by a surveillance weather radar using a NEXRAD 5-minute volumetric scanning strategy. This recommendation is not new. Based on experience and know limitations of volume scanning (lack of time resolution, blind cone), many campaigns have adopted additional research radars performing RHI scanning (eg. those cited in the manuscript, but also others, like LPVEX or IFLOODS) to track instrumented aircrafts, to analyze precipitating structure, or to obtain high resolution measurements along privileged direction, such that along instrumented sites. The step ahead is however the use a high-resolution simulator and a forward radar operator to quantify the advantage of RHI, depending also on the geometry of observations (eg the distance between radar and a convective cell).

Thank you for the referee's review and suggestions. We have referred to the studies listed above in the introduction and highlighted the advantage of the present study regarding the use of high resolution simulations in the abstract and introduction.

It is not clear to me, if, having at disposal one or two research radars, how the sector where RHI sweeps are performed, is identified. Typically, to this purpose, data from volume scans of an operational radar are used by an operator and likely optimal sector is subject to varying in time. Are tools like tobac helpful for an operator? Is it possible to switch to an unsupervised scanning? I think highlighting these points will improve the significance of the manuscript.

Thank you for giving a practical suggestion. In the present study, tobac was applied as a post processing. In the simulated scan strategies, tracking cells was guided by tobac using the VIL estimate from the model full grid every 1 minute. We have clearly mentioned this in Sect. 2.4.2 in the revised manuscript. For the real observations, to guide the RHI tracking for a convective cell, our group has developed an Multi Sensor Agile Adaptive Sampling (MAAS) flamework based on Kollias et al. (2020). The new MAAS has incorporated a cell-tracking algorithm using a watershed technique and predicts the future location of convective cells using multi sensors (e.g., satellite). This has been applied to the TRACER field campaign using the NEXRAD radar at Houston, TX and GOES-16 and showed good performance. We are preparing a separate paper focusing on the new MAAS flamework. In the revised paper, we have mentioned the MAAS in the summary section.

Specific comments.
L 54. Past experiences with PAWR should be better cited.

We have cited a few examples of studies using phased array radar systems (Billam and Harvey 1987; Heinselman and Torres 2011; Mahre et al. 2018; Griffin et al. 2019; Adachi and Mashiko, 2020; Moroda et al. 2021) in the introduction.

L 81. Spatial resolution of CR-SIM is not mentioned.

CRSIM outputs the same grid as the input model grid. We have added the information to Sections 2.1 and 2.4-1).

L 94. Although described in a different paper, could authors explain which radar measurement errors are included in the simulator?

The simulator does not simulate measurement errors that the real radar observations may have (e.g., noise-related error, accuracy, nonuniform beam filling effect), but it accounts for minimum detectable reflectivity as a function of distance. The simulator rather includes uncertainties associated with assumptions in the simulator such as hydrometeor particle shape (aspect ratio).

L 258. "Large rain">Large raindrops

We have changed it to "Large Dm for raindrops."

L 382. Please explain IOP

We rephrase this to read "a short-term term intensive observation period where such special scan strategies are performed".

L 464. The data availability statement should be more specific about accessing data used by the authors.

We have uploaded the VIL product from the two different environment simulations that was used for the cell tracking in this study to Stony Brook University Academic Commons. The data link is (https://commons.library.stonybrook.edu/somasdata/16. We have included this in the revised manuscript.

L 606. Is not clear if tobac identifies splitting and merging and how they are considered in Fig. 2

tobac can identify multiple convective cores embedded in a larger precipitation region and track the multi cores individually. The history record of the trackings does not include whether the tracked cell (core) split or merge, but we can identify it by checking the NetCDF file of cell regions with cell ID (at each timestep). Given the isolated, scattered nature of convection and the limited shear in these case studies (See Figure 2a,b), we expect that the impacts of splits and mergers would be minimal. Future versions of tobac (V1.5) will incorporate mergers and splits, but this was not available for our analyses.

L 627. It is not clear why the peak of Zdr (around 22) is not reflected in any features of Kdp

The larger ZDR shown at the height of ~4 km at around 22:00 UTC could reflect a fewer number of large oblate raindrops. In this region, the total liquid water content is small (<0.5 g m$^{-3}$) compared to the other

regions. Because KDP is proportional to the water content, the KDP values are small. I have added this discussion to Sect. 3.2.

---

## Author Comment (AC2)

Responses to Referee #2

Our response is highlighted by blue below each Referee comment in black.

The manuscript presents a discussion about scan strategies during measurement campaigns, aimed at optimizing the sampling of convective storms and the retrieval of updrafts intensity. The paper is in general well written, with appropriate scientific background and clear illustrations. One concern I had reading this manuscript is that besides the main topic (scan strategies), there are a couple of side topics (use of VIL for tracking, polluted vs. unpolluted convection simulation) which distract from the main topic. In my opinion, the authors should try to focus on the main topic and remove other topics that are not strictly necessary to the scope, and being treated only superficially, are not scientifically meaningful. For example, the conclusion that cell tracking works better with the use of the VIL only relies on the fact that VIL-based tracking has the largest total number of cells detected (fig. 2). Anyway, a serious comparison of different variables used for cell tracking should consider more statistical indicators and real data, e.g. to evaluate realistically the impact of false alarms, etc. I understand the intention of discussing the polluted vs. unpolluted cases to highlight the need for accuracy in the retrieval of vertical velocity. But for the scope of scan strategies, it may be enough to briefly report the ranges of vertical velocities simulated in different environments and refer to a separate (future) work specifically devoted to this analysis.

Thank you for your insightful comments on our manuscript. We understand the referee's point of view.

Regarding the unpolluted (CLN) and polluted (POL) cases, we agree that we are just scratching the surface. However, the reason for including the CLN and POL cases is two-fold. First, we use it to help demonstrate that using VIL-based tracking is more sensitive to the difference in cell lifetimes between the CLN and POL simulations, and therefore, may be suitable for tracking isolated convective cells throughout their lifetimes and quantifying cell-lifetime statistics. For example, when comparing the CLN and POL cases, the POL case had fewer cells, and this difference was consistent between the three tracking parameters. However, when assessing the distribution of cell lifetimes, the VIL-based tracking resulted in the clearest distribution shift towards relatively fewer long-lived cells in POL. This signal was shown in 10-dBZ-based tracking, and unclear in 40-dBZ-based tracking. This may work for the cases where isolated cells dominate in the domain with less stratiform or mesoscale precipitation areas and then the features identified by tobac in the VIL field well represent individual clouds (i.e., a single detected feature rarely includes more than one cells). We have added this discussion to the manuscript. Second, the differences between the CLN and POL cases are relevant to the community focused on aerosol impacts on deep convection and the TRACER/ESCAPE field campaigns. We also state that these impacts of aerosol on deep convection will be explored more fully in a future manuscript.

In our study, we did test using VIL to other commonly-used, single-level, reflectivity-based tracking. While these tests may not be comprehensive, we believe that they are both useful to the radar-tracking community and for the TRACER/ESCAPE field campaign efforts. We have changed the title of the section 3.1 to "Evaluation of the tracking parameter" and revised the text.

We are currently working on checking performances of the cell tracking algorithms using several parameters for tracking using both longer-period model data and real observation data involving multiple cell tracking algorithms. These findings are to be reported elsewhere. In this study, we focus on the high-resolution model outputs for the OSSEs rather than real observational data so as to avoid unknown observational uncertainties in the simulations of scans and retrievals.

About the main topic, I confess that from the title I had higher expectations. I would have expected a discussion about a possible objective methodology to adaptively optimize the scan strategies of several radars to minimize some cost function (e.g., the RMSE of vertical velocity if the aim is to study the updrafts). Instead, the cases treated are quite specific and hardly applicable to the set up of a generic campaign.

We again agree with the referee that the configurations of the radars employed in this study are specific to observing isolated deep convection. We have added the following sentences to the abstract: "The radar simulation settings are based on the Tracking Aerosol Convection Interactions ExpeRiment / Experiment of Sea Breeze Convection, Aerosols, Precipitation and Environment field campaigns held in Houston, TX." and changed the title to "Optimizing Radar Scan Strategies for Tracking Isolated Deep Convection Using Observing System Simulation Experiments". We have also defined that TRACER and ESCAPE campaigns will focus on observing isolated deep convective storms with different aerosol environments in the introduction.

In fact, basically one specific cell, at a given distance, or anyway equidistant from several radars in a network (fig. 9) is considered. What about if the cell location is not equidistant from all the radars? The ranges from the individual radars will change, e.g. it will not be possible to sample the cell with just 14deg azimuth sector if the cell is closer. What would be convenient in this case? Increase the azimuthal spacing of the RHI scans or decrease the temporal sampling for example? Which radar should perform a volume scan, and which should do a RHI scan? I would have expected to find answers to this kind of questions.

The reviewer raises a good point about the equidistant location of the radars, and we appreciate her/his suggestion. If the cell location is not equidistant from all the radars, the updraft retrieval quality may change depending on the distance. To test the sensitivity of having radars that are not equidistant, we have conducted the same analysis shown in Fig. 9, except that we varied the location of one of the radars from 10 to 70 km with a 10 km increment (except 20 km) and kept the other two radars at a fixed distance of 20 km (blue dots in Fig. 9a). Here we select 20 km for a fixed distance, since it shows a better retrieval result for the equidistance simulations. Figures R1a and R1b below show the vertical profiles of errors of the retrieved updrafts similar to Fig. 9b when moving the 2-min SEC radar (at the northwest corner of the triangle) and the 5-min VCP radar (at the south corner), respectively. When moving the 2-min SEC radar from the distance = 30 to 70 km (Fig. R1a), the retrievals show better profiles as the RMSEs range from 1.7 to 2.6 m s$^{-1}$. The RMSE increases when the radar is located at 10 km. This is consistent with the equidistance simulations (Fig. 9). Another notable point is that when the radar is located at 50 -70 km the errors below 1 km increase, likely because the radar coverage is sparse at the lowest elevation due to the distance. Similarly, when moving the 5-min VCP radar (Fig. R1b), the RMSE increases when the radar is located at 10 km. The errors are largest above 5 km altitude. When the radar is located at 60 or 70 km, as well as 10 km, the errors below 5 km increase. This also reflects the sparse radar coverage at the lower altitudes at the far distances. We have added this figure and the associated discussion to Sec. 3.3 in the revised manuscript and believe that it has made the manuscript more applicable to other studies. We also tested to vary the location of the third radar (at the northeast corner of the triangle), and the result is similar to Fig. 9a.

[Figure]

Fig. R1: Vertical profiles of errors of simulated retrievals (retrieval - model) averaged over a region with reflectivity > 30 dBZ at 21:42 UTC for the identified convective cell using the three radars where two are fixed at distance = 20 km (blue dots in Fig. 9a) while one is moved (dots with the other colors in Fig. 9a). In (a) the 2-min SEC radar (at the northwest corner of the triangle in Fig. 9a) is moved, and in (b) the 5-min VCP radar (at the south corner of the triangle in Fig. 9a) is moved. The colors of lines correspond to the locations of the moved radar in Fig. 9a.

The radar systems used for current field campaigns in conjunction with the operational radar network have more functionality (e.g., polarimetry, Doppler spectrum, and dual wavelength) and more configurable parameters (e.g., pulse sampling, pulse width, range-bin gate, azimuth, elevation spacings) than those used in the past. Therefore, the optimization of scan strategies for these more sophisticated radar systems is more complex. For more quantitative analyses, pre-field campaign OSSEs should also be conducted for each specific field campaign accounting for the characteristics of the radar systems that will be used for the field campaigns. Nevertheless, the radar specifications used in this study (e.g., azimuth spacing, beamwidth, elevation angles) are standard characteristics and have been widely used, and we argue that the results from this study can be qualitatively applied to other field campaigns that focus on deep convection. We have added this discussion to the summary section and revised the abstract and summary to mention the specific campaigns focused on there but have also noted the application of these approaches to other campaigns.

Having said that, the results presented are certainly useful for the specific set up of the planned measurement campaign in Texas. In this case, I recommend revising the paper (in particular, title/abstract) to make it clear that you're talking about a specific application. Otherwise, if the authors want to deal with the topic from a more general perspective (as the current title may suggest, at least to me), a major revision is needed, with a more comprehensive analysis of this (complex) problem and a clear organization of the material (also dropping all the unnecessary discussion about side topics).

As we responded to the previous comment, we have decided to change the title to be more specific and mention the specific campaigns focused on in this study in abstract and introduction. Thank you for the valuable suggestions and comments.

MINOR COMMENTS

Introduction: among previous work on scan strategy optimization, it would be worth adding at least a reference about the Collaborative and Adaptive Sensing of the Atmosphere (CASA) project, e.g.: Mclaughlin, David J., et al. "Distributed collaborative adaptive sensing (DCAS) for improved detection, understanding, and prediction of atmospheric hazards." Proc. American Meteorological Society Annual Meeting. 2005.

Thank you for introducing the work. I have referred to those works in the introduction.

Line 595: "the total beam" -> "the total number of beams"?

Yes, I have revised it.

Lines 151-159: it's not clear the difference and the goal of the scans named "1-min RHI" and "2-min SEC". Please explain better the difference between the two and why you need to split 0-45 and 45-90 the single RHI scans. Lines 336-339 later do not help clarifying.

For mechanically scanning radars, 1-min RHI may not be feasible due to mechanical limitations (e.g., overhead time), and those radars may therefore need more time to complete the sector scans (discussed in the next paragraph). We wanted to account for the mechanically scanning radars for the observation, because TRACER and ESCAPE planned to use mechanically scanning radars. To construct the cells observed by the sector scan that takes 2 min, we use two consecutive model snapshots; the first snapshot at the earlier time is used to simulate the scan of angles from 0.5° to 44.5° over the elevation, and the other is used to simulate the scans for the other angles from 45.5° to 89.5° over the elevation. This 2-min SEC simulation is performed every 2 min. We have rephrased the second paragraph of Sect. 2.4.2 to better enhance this explanation.

Table 1: why do you consider a different beam width (0.9deg) for the volume scan compared to the RHI scans (1.0deg)? I suppose this is to emulate NEXRAD's VCP, but why it is necessary? Shouldn't the comparison be valid for a generic radar? From the title and abstract I though the study aimed at a generic theoretical evaluation of scan strategies, but from these settings it looks like the goal is more specific and concerns the combination of NEXRAD and ARM radars. This should be mentioned more clearly in the abstract/intro.

We wanted to use the radar configurations from real radars that are available for the actual field campaigns (i.e., TRACER and ESCAPE). The use of different beam widths for each radar is not a target of the OSSEs. The referee is right; this setting represents a specific radar (NEXRAD). However, we will also note that NEXRAD radars are available across the U.S, and many field campaigns have and will continue to use NEXRAD radars in their observational networks, and we believe that the configurations are generalizable for campaigns focused on isolated deep convection. We have revised the abstract and title to represent the aims of the present OSSEs and specify the cases (see the response to the general comments).

Figure 1 is mentioned for the first time after figure 8 (at line 302). Check figure order and corresponding references in the text.

We referred to Figs. 1a and 1b at the beginning of Sect. 2, Fig. 1c in Sect. 2.2, Fig. 1d in Sect. 2.4.1 in the original manuscript. We have added the following sentence to the beginning of Sect. 2: "Figures 1a-1c show example snapshots from the parts (1) and (2), and Figure 1c shows the tracking result from the part (3)."

Figure 3: units of VIL is kg/m2, but dB is used here. You may use a logarithmic scale for the x-axis, but keeping the correct measurement units.

We have changed the x-axis label to "$10 \log_{10}(\text{VIL } [\text{kg m}^{-2}])$".

Figure 4: what is the height of the freezing level for this case?

The freezing level (height at T=0°C) is approximately 4.7 km at 21:47 UTC. We have added a 0°C isotherm of domain averaged temperature to each panel in Figures 4 and 5.

Figure 5: I would expect to see some positive Zdr above the freezing level corresponding to the strongest updraft… Maybe the average over the 40 dBZ area masks the Zdr columns (if present)?

This convective cell produces a significant amount of hail, which is assumed to be dry and have near-spherical shapes (Ryzhkov et al. 2011) in the radar simulator, thereby producing small ZDR. This may mask the ZDR column which is attributed to (supercooled) raindrops in the total radar observables. We plot the height-versus-time cross sections of Zhh, ZDR, and KDP for rain scattering only from the same convective cell:

[Figure]

Figure R2: Time-height cross sections of (a) C-band radar reflectivity, (b) ZDR, and (c) KDP from rain scattering, averaged for areas with the reflectivity > 30 dBZ for the selected convective cell from the model simulation. The dashed line in each panel represents a 0°C isotherm of domain averaged temperature.

The values in the plot are the average for Zhh>30 dBZ. The positive ZDR (>1 dB) region extends to 6 km altitude at around 21:38 UTC, where large KDP is also shown. Looking at Figure 5, the ZDR extension is also clearly evident in Figs. 5d and 5e, but it is not clear or is weak in Fig. 5f. Vertical cross sections of the cell at 21:38 UTC shows the ZDR column clearly:

[Figure]

Figures R3: Vertical cross section of C-band total (a) radar reflectivity, (b) ZDR, and (c) KDP from all hydrometeor scatterings for the selected convective cell at 21:38 UTC. Dashed line in each panel represents a 0°C isotherm.

The ZDR column (and KDP column also) extends to 5.8 km altitude, which is approximately 1 km above the environmental 0ºC level level.

We have added this discussion to Sect. 3.2. Thank you for pointing this out.

Figure 10: the meaning of the colored arrows (and why they are repeated at different ranges) should be explained.

The arrows in Fig.10b represent examples of the Doppler velocity vector and the components at two range-height bins ([x,y]=[0 km, 5 km] and [7.5 km, 5 km]); the clear arrows with black line represent the observed Doppler velocity (labeled as DV), the red arrows represent vertical velocity component (labeled as VV), the yellow arrows represent horizontal wind component along the vertical cross section (labeled as U), and the light blue arrows represent the component of the particle fall velocity (labeled as Vfall). We have added the information in the caption of Fig. 11. These examples assume that each component at the two points has the same value, but the scale does not represent a specific value. At the radar distance is equal to 0 km (x=0 km), the horizontal wind component can be ignored. At the radar distance is not equal to 0 km, the contribution of the horizontal wind component increases with decreasing the elevation angle (i.e. increasing the distance from the radar at a constant height). This explanation has been added to Sect. 3.3.